# Response of the sea surface temperature to heatwaves during the France 2022 meteorological summer

Thibault Guinaldo [1], Aurore Voldoire [2], Robin Waldman [2], Stéphane Saux Picart [1], and Hervé Roquet [3]

[1]Centre National de Recherches Météorologiques, Université de Toulouse, Météo-France, CNRS, Lannion, France
[2]Centre National de Recherches Météorologiques, Université de Toulouse, Météo-France, CNRS, Toulouse, France
[3]Direction de l'enseignement supérieur et de la recherche, Météo-France, Saint Mandé, France

**Correspondence:** Thibault Guinaldo (thibault.guinaldo@meteo.fr)

**Abstract.** Summer 2022 was a memorable and record-breaking event, ranking as the second hottest summer in France since 1900, with a seasonal surface air temperature average of 22.7°C. In particular, France experienced multiple record-breaking heatwaves during the meteorological summer. As the main heat reservoir of the Earth system, the oceans are at the forefront of events of this magnitude which enhance oceanic disturbances such as marine heatwaves. In this study, we investigate the sea surface temperature (SST, hereafter) of French maritime basins using remotely-sensed measurements to track the response of surface waters to the atmospheric heatwaves and determine the intensity of such feedback. Beyond the direct relationship between SSTs and surface air temperatures, we explore the leading atmospheric parameters affecting the upper-layer ocean heat budget.

Despite some gaps in data availability, the measured SSTs during the meteorological summer of 2022 was record-breaking, the mean SST was between 1.3 °C and 2.6 °C above the long-term average (1982-2011) and studied areas experienced between 4 and 22 days with average SSTs beyond the climatological maximum. We find out a significant SST response during heatwave periods. with maximum temperatures locally measured at 30.8 °C in the northwestern Mediterranean Sea. Our results show that France experienced in August 2022 (July 31st to August 13th) above-average surface solar radiation correlated with below-average total cloud cover and negative wind speed anomalies. Our attribution analysis based on a simplified mixed layer heat budget highlights the critical role of ocean-atmosphere fluxes in initiating abnormally warm SSTs while ocean mixing plays a crucial role in the cessation of such periods. We find that the 2-m temperatures and specific humidity are key variables across all regions studied that are consistently linked to the advection of warm and moist air masses. Our results reveal that the influence of wind on heatwaves is variable and of secondary importance. Moreover, we observe that the incident solar radiation has a significant effect only on BB and EC areas. Our results are in line with previous studies, and demonstrate that even if the Mediterranean is known as a climate change hotspot, all the studied maritime areas are affected by a continuous warming of surface water and responded to extreme synoptic conditions.

Our study therefore provide valuable insights into the complex mechanisms underlying the ocean-atmosphere interaction and demonstrates the need for an efficient and sustainable operational system combining polar orbiting and geostationary satellites to monitor the alterations that threaten the oceans in the context of climate change.

*Copyright statement.*  TEXT

## 1   Introduction

Anthropogenic influence on the climate is unequivocal and leads to a global Earth's energy imbalance, as evidenced by changes in global average atmospheric temperature distributions (Eyring et al., 2021). Furthermore, anthropogenic global warming is impacting climate variability, resulting in an increase in both the intensity and the frequency of extreme events (IPCC, 2021;
Li et al., 2021). Regional projections, combining regional climate projections and historical observational constraints, predict warming in France to range between 2.9°C and 4.8°C by 2100 in a medium emission scenario, with an increased impact on summer temperatures (Ribes et al., 2022). These extremes, which affect every component of the Earth's system, are primarily driven by atmospheric synoptic circulation (Trenberth et al., 2015; Faranda et al., 2020).

The ocean plays a crucial role in the Earth's energy balance, with its extensive volume and inherent thermal capacity making
its contribution greater than that of any other component (continents, atmosphere, glaciers). In fact, the oceans absorb more than 90% of the thermal excess, leading to an increase in global ocean heat content and temperatures (Cheng et al., 2019). This internal energy imbalance in the ocean has direct consequences on the properties of the system, its dynamics, its contribution to the water cycle, and its unique biotope. Such changes in global ocean heat content have direct impacts on sea-level rise (Cazenave et al., 2018), the total water vapor column (Trenberth and Shea, 2005), melting of continental glacier platforms
(Rignot et al., 2014) and decline in ocean dissolved oxygen (Keeling et al., 2010). The ocean's tremendous heat capacity mitigates the surface atmospheric warming, which is 50% higher over land than above the ocean (IPCC, 2021). Among other oceanic events influenced by the context of climate change, marine heatwaves (hereafter MHWs), which are defined as events of anomalously warm water temperature with specific characteristics (spatial extent, duration, intensity), have emerged as a key field of research due to their substantial impact on ocean ecosystems (Hobday et al., 2016; Sen Gupta et al., 2020). Although
the extent of such events is not limited to the ocean surface (Schaeffer and Roughan, 2017)), MHWs are commonly detected and identified through sea surface temperature (SST) measurements (Oliver et al., 2021; Benthuysen et al., 2018) and the shift in their occurrence is directly linked to SST trends (Sen Gupta et al., 2020).

SST is a fundamental variable in oceanographic monitoring and forecasting systems on both daily and climatic timescales. SSTs are also key indicators of global ocean warming, which have increased since the beginning of the 20th century and are
expected to continue on this pathway due to the direct influence of near-surface atmospheric forcing changes (IPCC, 2019). These shifts in SSTs also have implications in return, several studies demonstrating the contribution of warm SST anomalies in the enhancement of atmospheric heatwaves (Feudale and Shukla, 2007; Mecking et al., 2019; Hong et al., 2021). However, the operational and climatic needs for observations require a spatial and temporal coverage that cannot be achieved by *in situ* measurements alone. Satellite SST measurements, due to their global spatial coverage and usable spatial and temporal resolu-
tion, meet these operational and climatic requirements and are now integrated into forecasting systems (Donlon et al., 2012; Minnett et al., 2019; O'Carroll et al., 2019).

This study focuses on two main objectives. First, we assess and characterise the response of SSTs over the French sea basins to the atmospheric heatwaves that hit France during the meteorological summer of 2022 using remotely-sensed SST observations. Finally, we diagnose the influence of atmospheric variables on the SST response and the relations between these variables based on a simplified mixed layer heat budget. We disentangle the role of the different atmospheric variables in explaining the anomalies in the surface flux and therefore their respective role in driving the mixed layer warming. The objective is to gain a better understanding of the relationship between atmospheric conditions during the meteorological summer of 2022 and the underlying SSTs.

Section 2 provides a comprehensive overview of the data and methods used in this study. The synoptic conditions that triggered the heatwaves in Western Europe during the summer of 2022 are described in detail in Section 3. The main results of the study, including the characterization of the response of SST and its relationship with the physical processes at the air-sea interface, are presented in Section 4. Finally, in Section 5, we discuss these results, draw conclusions on their implications and the plan for future research.

## 2 Data & Methods

### 2.1 Study sites

In this study, we focus on the responses to heatwaves that affected France during the meteorological summer of 2022. However, it is important to note that meteorological extreme events such as heatwaves have a spatial extent beyond land borders. In light of this, we have selected three distinct oceanic regions, including both coastal and open ocean areas, as presented in Figure 4. These regions are characterized by distinct features and are as follows:

– **Area EC**: The English Channel, characterized by strong tidal currents that enhance water column mixing, thereby helping to maintain cooler SSTs. This area is directly affected by the atmospheric zonal flux.

– **Area BB**: The Bay of Biscay which is part of the Atlantic ocean and is composed of a deep abyssal plain connected to a shallow continental shelf by steep continental slopes. This area is also directly affected by the atmospheric zonal flux.

– **Area NWM**: The North-Western Mediterranean Sea sub-basin, including the Gulf of Lions, the Ligurian Sea and the Balearic basin which are in the Copernicus Marine Environment Monitoring Service (CMEMS) Mediterranean Sea validation procedure as mentioned in Lazzari et al. (2021). This area is directly influenced by Mistral and Tramontane regional winds which drive recurrent upwelling phenomena, making it of particular interest in comprehensive studies of the Mediterranean water cycle and its implications for climate studies (Drobinski et al., 2014; Ruti et al., 2016).

### 2.2 Atmospheric reanalysis

The overview of the exceptional weather situation in France during the summer of 2022 is based on the systematic analysis of different meteorological variables influencing the upper-ocean energy balance (Equation 1). The forcing data used in the mixed

layer heat budget were from the ERA5 reanalysis regridded to a regular latitude-longitude grid of 0.25 degrees (Hersbach et al., 2020).

We also used an atmospheric forcing climatology to test the heat flux sensitivity the each variable in the model experiments. In contrast to the SST products, we used a 1991-2020 time period reference, in accordance to the World Meteorological Organisation standards (hereafter WMO). Both hourly and monthly ERA5 data were used, the first to compute daily and weekly anomalies while the second were used to compute the 1991-2020 monthly climatological mean.

## 2.3 Sea surface temperature data

### 2.3.1 Operationnal SST product

The Ocean and Sea Ice Satellite Application Facility (OSI SAF) provides users with operational products of SST in near real time (https://osi-saf.eumetsat.int/). In particular it has been delivering operational Metop/Advanced Very High Resolution Radiometer (AVHRR) SST products since 2007. In this study the specific OSI SAF product used is the level-3C (mono-sensor collated) from Metop-B labelled as OSI-201-b (https://osi-saf.eumetsat.int/products/osi-201-b). It is a global $0.05°$ gridded product available twice daily.

SST retrieval from AVHRR data relies on MAIA version 4 cloud mask (NWP SAF, 2017) and is based on a split-window algorithm using two infra-red bands at 10.8 and 12.0 $\mu$m which coefficients are tuned so that the retrieved SST has a global zero bias against drifting buoy measurements at about 20cm depth, see for example Marsouin et al. (2015). Since 2017, operational production of Metop SST includes an algorithm correction scheme. This scheme has been designed to mitigate the SST algorithm inherent biases due to changing atmospheric conditions (Le Borgne et al., 2011; OSI SAF, 2018b).

As the product is operational, it is necessary to be aware of the biases inherent in the data. In order not to introduce biases related to the diurnal cycle, only night-time data have been analysed. However, these data are aggregated over a time window of 12 hours centered at midnight UTC. For the studied area, the night-time data are acquired at 21:30 local time, thus in the westernmost part of the domain data are acquired at dusk in summer time. Therefore, data with a solar zenith angle smaller than 95° are removed to avoid analysing daytime SST. OSI SAF SST products are delivered along with a per-pixel Quality Level which reflects the quality of the retrieval. This includes considerations about potential contamination by cloud and mineral dust aerosols. The QL is ranging from 0 (no data) to 5 (best available quality).

### 2.3.2 ESA CCI SST product

For the purpose of climatological computations, the version 2.1 of the European Space Agency Climate Change Initiative (ESA CCI) level 4 Climate Data Record (hereafter CDR) were used (Good et al., 2019; Merchant et al., 2019). These SST CDR measurements are based on the cloud-free reprocessed thermal infrared radiance from the AVHRR and the Along Track Scanning Radiometer (ATSR) sensors. It is a daily global gap-free product available from 1981 to 2016 on a regular latitude/longitude grid at 0.05 degrees resolution. This analysis represents the daily mean 20cm depth SST corrected by the diurnal cycle. Thus, this measurement is equivalent to the nighttime SST products developed under the OSI SAF framework.

### 2.3.3 SST climatology and anomaly

In the present study, we calculated both daily and monthly SST anomalies. To construct a reliable climatology for SST anomalies, we used a 30-year archive of data specifically dedicated to climate studies, in accordance to the standards set by the WMO. We obtained monthly climatological averages by averaging daily ESA CCI CDR (Section 2.3.2) using the Climate Data Operator tool (Schulzweida, 2022) for the 1982-2011 period. We then calculated daily (resp. monthly) anomalies by comparing the constructed daily (resp. monthly) ESA CCI climatology to the daily (resp. monthly) OSI SAF SST data.

To reduce biases in the analysis, we filtered the operational OSI SAF SST data by only using data with a Quality Level parameter of 3, 4, or 5, as recommended in the Product User Manual (OSI SAF, 2018a). Additionally, we ensured that the number of measured pixels was representative of the area by only keeping days where measurements covered at least 50% of the total surface area. To limit disturbances in the SST analysis, we applied a sliding window over 3 days and considered only lower frequencies, which also reduced the noise introduced by the diurnal cycle.

We also note that no changes were made to the SST retrieval process during the study period, ensuring that the SST are homogeneously distributed in both space and time.

### 2.4 Modeling framework

The aim of this research is to gain a deeper understanding of how SSTs react to the abnormal conditions of the summer of 2022 through examination of ocean-atmosphere interactions. We investigate the relationship between SSTs and specific atmospheric conditions, as well as the contribution of the inherent atmospheric variables. Generally, changes in SSTs are primarily the result of small-scale processes occurring within the mixed layer that can be enhanced by climate modes such as the NAO (Holbrook et al., 2019). Among these local processes, Chen et al. (2014) found that the effects of a northward shift in the jet stream on SSTs are primarily driven by changes in the net heat flux at the ocean-atmosphere interface. Hence understanding the generation of warm SST conditions needs to be addressed by studying the atmospheric interactions with the ocean mixed layer.

#### 2.4.1 Mixed layer heat budget

A mathematical approach to the mixed layer heat budget had been proposed by Moisan and Niiler (1998) under the Boussinesq approximations, Reynolds averaging and diffusive closure assumptions for turbulent fluxes:

$$\frac{\partial T_m}{\partial t} = \underbrace{-\overline{\mathbf{u}} \cdot \nabla T_m}_{A} + \underbrace{\kappa_h \Delta T_m}_{B} \underbrace{- \frac{1}{h} \left[ \kappa_v \frac{\partial T}{\partial z} \right]_{-h}}_{C} - \underbrace{\left( \frac{T_m - T_{-h}}{h} \right) \left[ \frac{\partial h}{\partial t} + \mathbf{u}_{-h} \cdot \nabla h + w_{-h} \right]}_{D} + \underbrace{\frac{q_{rad} + q_{turb}}{\rho_0 c_w h}}_{E} \tag{1}$$

where $T_m$, $h$, $\rho_0$ and $c_w$ are respectively the mean temperature, the depth, the mean density and the specific heat capacity of the surface mixed layer, $\mathbf{u}$ is the horizontal velocity vector and $w$ is the vertical velocity. $\kappa_h$ and $\kappa_v$ are respectively the horizontal

and vertical turbulent diffusivity coefficients. $q_{rad}$ is the net radiative heat flux and $q_{turb}$ is the net turbulent heat flux at the sea surface, removing from the former the fraction that is radiated below the mixed layer. The mention $-h$ refers to the bottom of the mixed layer as the vertical axis is oriented upward.

This equation helps to understand the contribution of each process to the mixed layer heat budget. The SST tendency is dependent on the horizontal advection (A), the horizontal eddy transport (B), the vertical turbulent mixing (C) and entrainment (D) of heat at the mixed layer base and the ocean-atmosphere interface heat flux (E). One can notice the contributions of the solar shortwave radiation (through component E) and the wind speed through ocean currents (components A and D), vertical turbulence (component C) and air-sea heat fluxes (component E) on the SST evolution.

In this study, we aim at relating air-sea heat fluxes to the ocean mixed layer temperature trend. Therefore we write a simplified form of Equation 1 as:

$$\frac{\partial T_m}{\partial t} = \underbrace{\frac{q_{rad} + q_{turb}}{\rho_0 c_w h}}_{E} + \underbrace{Res}_{A+B+C+D} \tag{2}$$

where only the air-sea heat flux (component E) is explicitly diagnosed, the remaining terms being merged into a residual $Res$ deduced from the difference between the total trend and the air-sea heat flux trend. We interpret the latter term as a cooling trend mainly driven by vertical turbulent exchanges (namely, components C and D), consistently with the literature on marine heatwave heat budgets (e.g. Amaya et al. (2020)).

Radiative air-sea heat fluxes are composed of shortwave incoming solar radiation ($Q_{SWD}$), shortwave radiation reflected at the surface ($Q_{SWU}$), longwave downward radiation ($Q_{LWD}$) and the longwave upward ($Q_{LWU}$) contribution from the ocean. Net radiative fluxes over the ocean can then be summarized as the sum of the net shortwave radiation ($Q_{SW}$) and the net longwave radiation ($Q_{LW}$).

Turbulent fluxes are composed of sensible and latent heat fluxes which are estimated, following the Monin-Obukhov Similarity Theory (MOST, Monin and Obukhov, 1954), by the so-called Bulk aerodynamic formulas:

$$Q_S = \rho_0 C_p C_S |U_{10m}| \Delta T \tag{3}$$

$$Q_L = \rho_0 L_v C_L |U_{10m}| \Delta q \tag{4}$$

with $C_p$ the air specific heat capacity, $C_S$ and $C_L$ respectively the sensible and latent heat transfer coefficients, $L_v$ = 2.26 $MJ/kg$ the latent heat of evaporation, $U_{10}$ the wind speed at $10m$, $\Delta T$ the ocean-atmosphere interface temperature difference ($K$), $\Delta q$ the ocean-atmosphere interface specific humidity difference ($g.kg^{-1}$). The specific humidity at the ocean surface is given by:

$$q_s = 0.98 q^*(SST) \tag{5}$$

with $q^*(SST)$ the saturating humidity at the sea surface temperature.

Under this decomposition, equation 1 becomes:

$$\frac{\partial T_m}{\partial t} = \frac{1}{\rho_0 c_o h}\Big(Q_{SW}(0) - Q_{SW}(-h) + Q_{LW} + Q_S + Q_L\Big) + Res \tag{6}$$

### 2.4.2 Air-sea heat flux computation

To be able to assess the role of the different atmospheric variables in driving the net atmospheric fluxes and thereby the SST evolution, we have used the surface modeling platform SURFEX (VERSION 8) (Masson et al., 2013; Le Moigne et al.,2020),

developed at CNRM to calculate turbulent fluxes and radiative upward fluxes depending on incoming radiation and atmospheric variables. Turbulent fluxes are estimated using the COARE version 3.0 bulk formulae (Fairall et al., 2003), radiative upward fluxes are calculated considering an ocean albedo of 0.065 and an emissivity of 0.96 respectively for shortwave and longwave radiation.

### 2.4.3 Mixed layer depth reconstruction

The mixed layer depth (MLD hereafter) is an oceanic variable that influences upper ocean variability and controls biogeochemical processes. While this variable is not specifically analyzed in the present study, it is necessary for the computation of the mixed layer heat budget. Therefore, we used the CMEMS mixed layer depth analysis for the summer of 2022 to compute an average mixed layer depth. Specifically, this value is the temporal average of respectively the hourly CMEMS Mediterranean

Forecasting System (Med-Physics) for the NWM area and the CO5 configuration of the Atlantic Margin Model (O'Dea et al., 2017) over the summer of 2022 for EC and BB. Over the regions of interest, this averaged MLD is deeper than 10m (Figure A2), a depth for which typically 90% of the incoming solar radiation has been absorbed. therefore, for simplicity, we assume that the mixed layer absorbs all the surface solar radiation (hence $Q_{SW}(-h) \simeq 0$). Considering the low variability of the MLD (Figure A4) throughout the summer, we used a mean MLD of 12.5m for the NWM region and 11m for the BB region in our

study. The outcome is different for the EC region with a strong spatial and temporal variability throughout the summer. For this specific area, we decided to prescribe the daily MLD directly in the analysis.

### 2.4.4 Model sensitivity experiments

Our simple model finally consists in estimating the air-sea fluxes depending on the atmospheric state using SURFEX and then estimating the SST evolution depending on these fluxes using the simple ocean bulk temperature equation (Eq. 6). We

used SURFEX in offline mode utilizing ERA5 atmospheric variables as forcing data on the ERA5 grid. In all experiments,

fluxes are calculated at an hourly time-step over the three months considered in the study using ERA5 hourly data. A reference experiment (CTL hereafter) is conducted in which all atmospheric forcings are prescribed according to their value in 2022.

To diagnose the influence of atmospheric variables on the SST response, an experiment in which all atmospheric parameters are modified to their values of the period 1991-2020 has been conducted (CLIM hereafter). This is done by estimating the flux taking atmospheric parameters from each summer of the period and then averaging the fluxes obtained over the full period. This gives an estimation of "climatological fluxes", assuming an unchanged SST. Integration of the ocean bulk model driven by these climatological fluxes provides a reference SST evolution that would happen if atmospheric parameters where climatological (CLIM). Based on this CLIM experiment, a set of sensitivity experiments has been conducted in which atmospheric parameters are modified individually to their 2022 value (as in CTL). All atmospheric variables used as inputs in the net heat flux equation have been tested : temperature at 2 m (T2M hereafter), specific humidity at 2 m (HUS hereafter), 10 m wind speed module (WND hereafter), and incoming shortwave (RSS hereafter) and longwave (RLS hereafter) radiations. Experiments and their acronyms are listed in Table 1. Note that the air-sea flux computation also depends on SSTs. Here, we keep the SST as observed in 2022. The model set-up thus does not consider the feedback effect of SST on air-sea fluxes, which is out of the scope of this study.

The contribution of each variable on SST was then determined by calculating the difference between SSTs of the respective sensitivity experiment and the CLIM experiment, similarly, the effect of all atmospheric variables is calculated as the difference CTL - CLIM.

## 3 Synoptic conditions initiating heatwaves during the summer of 2022

During the meteorological summer of 2022 (June-July-August), France experienced the second warmest summer since 1900 with surface air temperatures that broke records in terms of both intensity and earliness (seasonal surface temperature average anomaly of +2,3°C). The summer also set a record for the number of days spent under heatwaves, with 33 days split across three distinct events. The associated synoptic conditions resulted in anomalies in most of the variables influencing air-sea heat

**Table 1.** List of model experiments with their respective forcing. CLIM refers to the average reference period 1991–2020.

| Experiment name | CTL | CLIM | T2M | HUS | WND | RSS | RLS |
|---|---|---|---|---|---|---|---|
| Temperature at 2 m | 2022 | CLIM | 2022 | CLIM | CLIM | CLIM | CLIM |
| Specific humidity at 2 m and surface pressure | 2022 | CLIM | CLIM | 2022 | CLIM | CLIM | CLIM |
| Wind speed at 10 m | 2022 | CLIM | CLIM | CLIM | 2022 | CLIM | CLIM |
| Incoming shortwave radiation | 2022 | CLIM | CLIM | CLIM | CLIM | 2022 | CLIM |
| Incoming longwave radiation | 2022 | CLIM | CLIM | CLIM | CLIM | CLIM | 2022 |

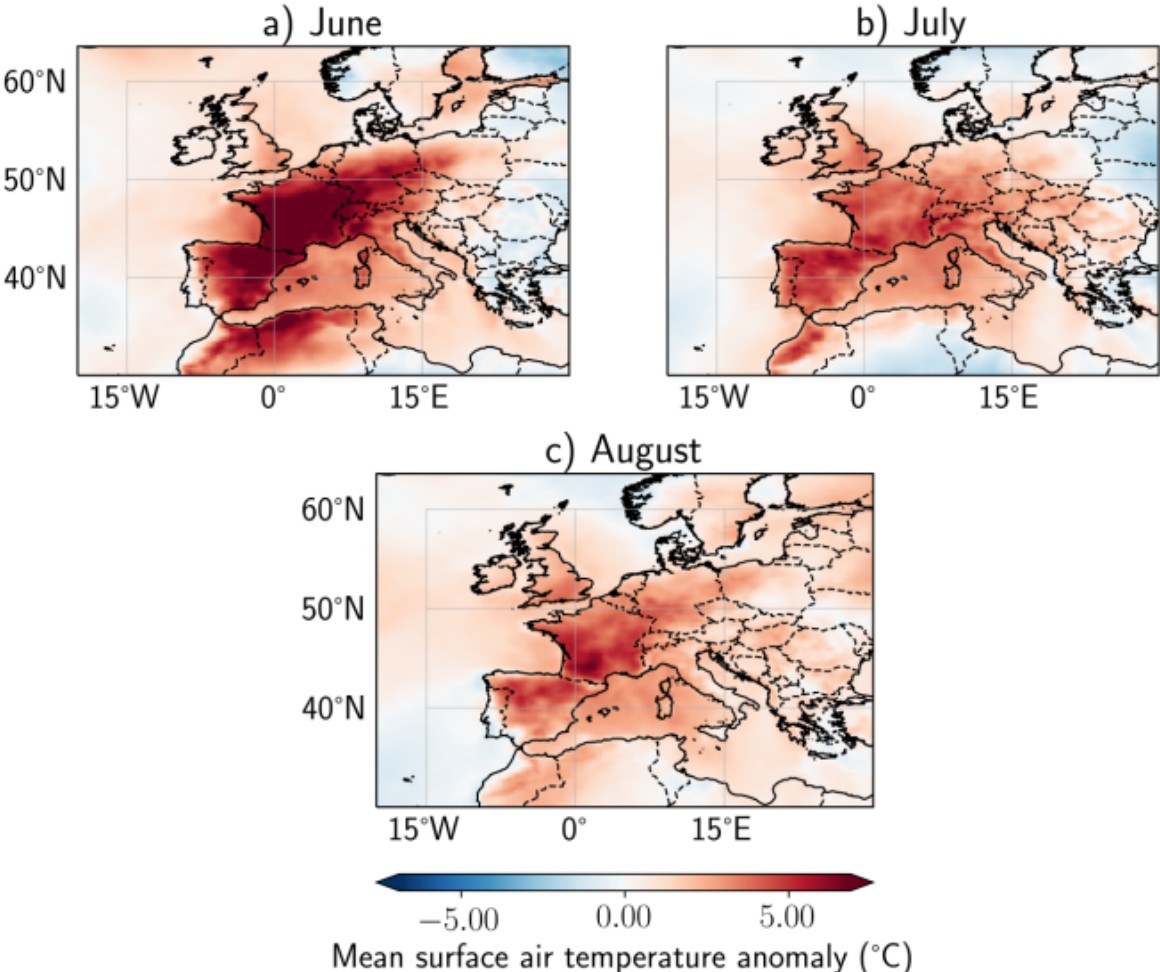

**Figure 1.** ERA5 reanalysis of mean temperature anomaly at 2m height for the a) June 15th-19th , b) July 11th-25th and c) July 31th - August 13th 2022 heatwaves compared to the corresponding period of the 1991-2020 climatology.

fluxes. Among them, the 2m air temperature reached anomalies exceeding 5°C over western Europe (Figure1). Particularly, all of the metropolitan French territory and its surrounding sea basins have been affected (Figure1).

The first heatwave occurred between June 15th and 19th, and was caused by a shift in the weather pattern from a zonal regime to a summer blocking. This led to a stationary north-south meander of the jet stream and the formation of a cut-off low (hereafter CUL) over the Iberic Peninsula. This low-pressure system brought southerly winds that advected hot air masses, known as heat plumes, over western Europe and France as seen in the Figure 2 representing the anomaly in the 500 hPa geopotential height. The second heatwave, which occurred between July 11th and 25th, and the third heatwave, which occurred between July 31st and August 13th, both had a similar dynamic of formation. They were linked to a north-south planetary wave swell that caused

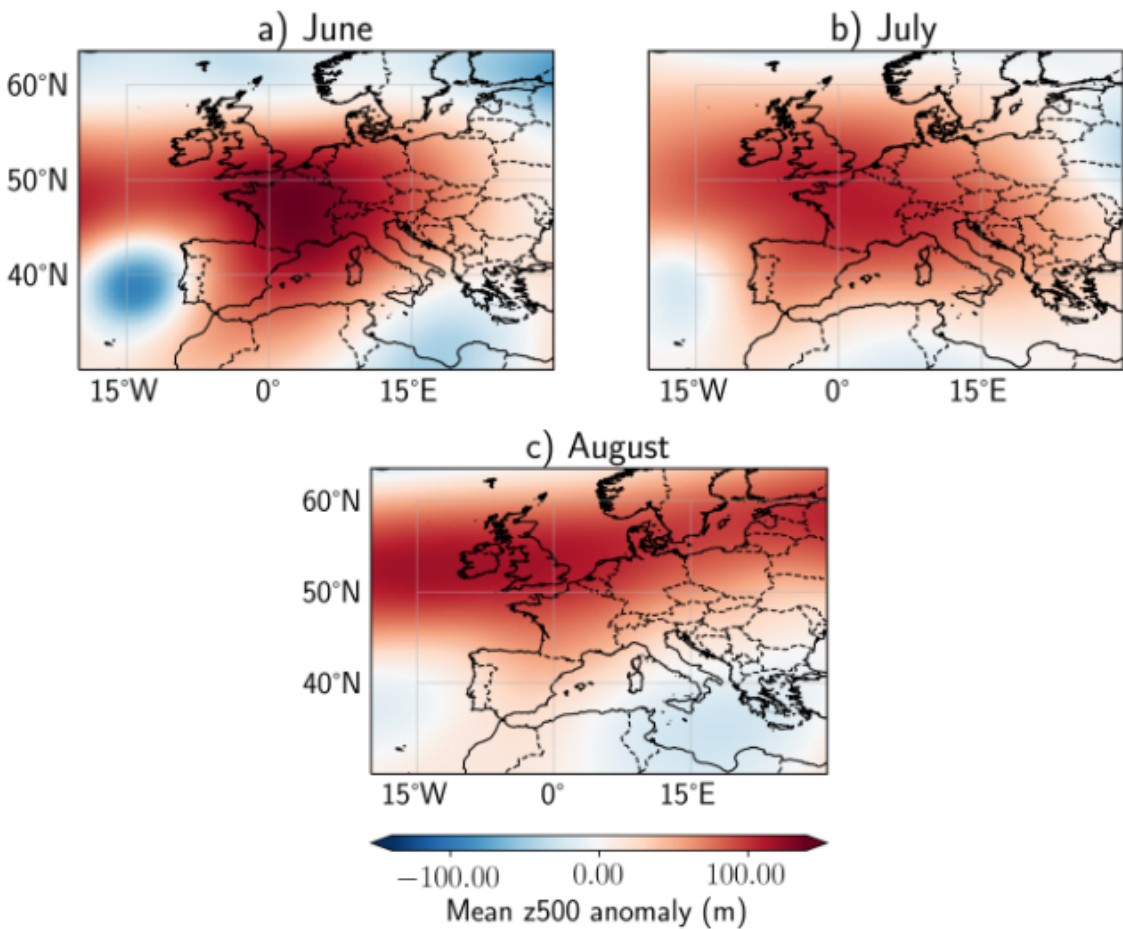

**Figure 2.** Same as Figure1 for the mean geopotential height at 500 hPa pressure level anomaly for the a) June, b) July and c) August heatwaves compared to the corresponding period of the 1991-2020 climatology.

a meander of the jet stream. This led to adiabatic compression of subsiding air masses, which created a heat dome over the

245 western part of Europe. In addition, the meander of the jet stream triggered the formation of another CUL, which further enhanced the already abnormally warm surface air temperatures by advecting hot air masses from the Sahara. In all of the three heatwaves, the air mass advection occurred over the Mediterranean Sea or the Bay of Biscay which moistened the air mass which hence increased the specific humidity and had a strong effect on surface temperatures (Santos et al., 2015). These conditions have been well studied in previous research which indicates that these dynamics are usual in the occurrence of

250 heatwaves over western Europe (Zschenderlein et al., 2019).

# 4 Results

## 4.1 Daily sea surface temperature evolution over the 2022 meteorological summer

The primary objective of an operational product is to provide daily monitoring for use by forecasting services. However, basin-scale global coverage is dependent on several conditions described in the Section 2.3.3 which are not always met in all basins on a daily basis. As a result, a significant portion of the data may not be available due to various factors such as clouds, aerosols, and low quality data. The extent of missing daily SST data varies for each basin as illustrated in Figure 3. One can note that the EC, with 60% of missing data, is more impacted than the BB (48%) and the NWM (29%) areas where the missing data is significant only until the beginning of July. These missing data do not follow a regular pattern and are rather related to the applied filtering conditions. In particular, the missing data does not permit a systematic analysis of the response of the three basins to the early heatwave in June 2022 (June 15th to 19th) and the July 2022 (July 11th to 25th) heatwave. However, the response of the July 2022 heatwave is conceivable for the BB and NWM areas. Only the response to the August 2022 heatwave is feasible for all three basins. Therefore, the analysis of the SST response to heatwaves in this study focuses on the August 2022 event.

Despite this limitation, it is still possible to have an overview of the exceptional summer of 2022. As seen in Table 2 and Figure 3, all three basins experienced a range of warmer-than-average SSTs, with the strongest response seen in the NWM area (mean daily anomaly of 2.6 °C and maximum of 4.3°C). Notably, there were no days at the basin scale where temperatures were within the normal temperatures range or below. The summer of 2022 also a record for this basin with an average temperature of 26.1 °C. Over the period of 1982-2011, the previous record dated back to 2003 with 25.6 °C. Furthermore, the response of the NWM basin is clear with an increase of the average SSTs of 2.7°C between July 6th and July 20th (during which the basin experienced 12 continuous days with an average anomaly over 3°C) and of 1.4°C between July 31st and August 11th (during which the basin experienced 9 continuous days with an average anomaly over 3°C). Based on the available data, they were no days where SSTs were close or below the climatology on the EC and BB regions. 1.4°C and 1.3°C for EC and BB respectively, these regions experienced a warmer than usual summer. The BB basin showed a less pronounced response to the August heatwave, resulting in a comparatively lower magnitude of warming.

Nonetheless, the surface ocean of all three regions was therefore anomalously warm throughout the summer. This is also highlighted by the variability, presented in the Table 2, which is comprised between 31% and 46% of the mean SST anomaly for a standard deviation between 0.5°C and 0.8°C. With the exception of specific episodes, SSTs remain close to the climatological maximum of the period 1982-2011 (Figure3). In addition, it is noteworthy that the NWM experienced 22 days, EC experienced 19 days, and BB experienced 4 days of SSTs exceeding the climatological maximum. It should be noted that the previous temperature record in the NWM dated back to 2003, underscoring the historical significance of the observed response.

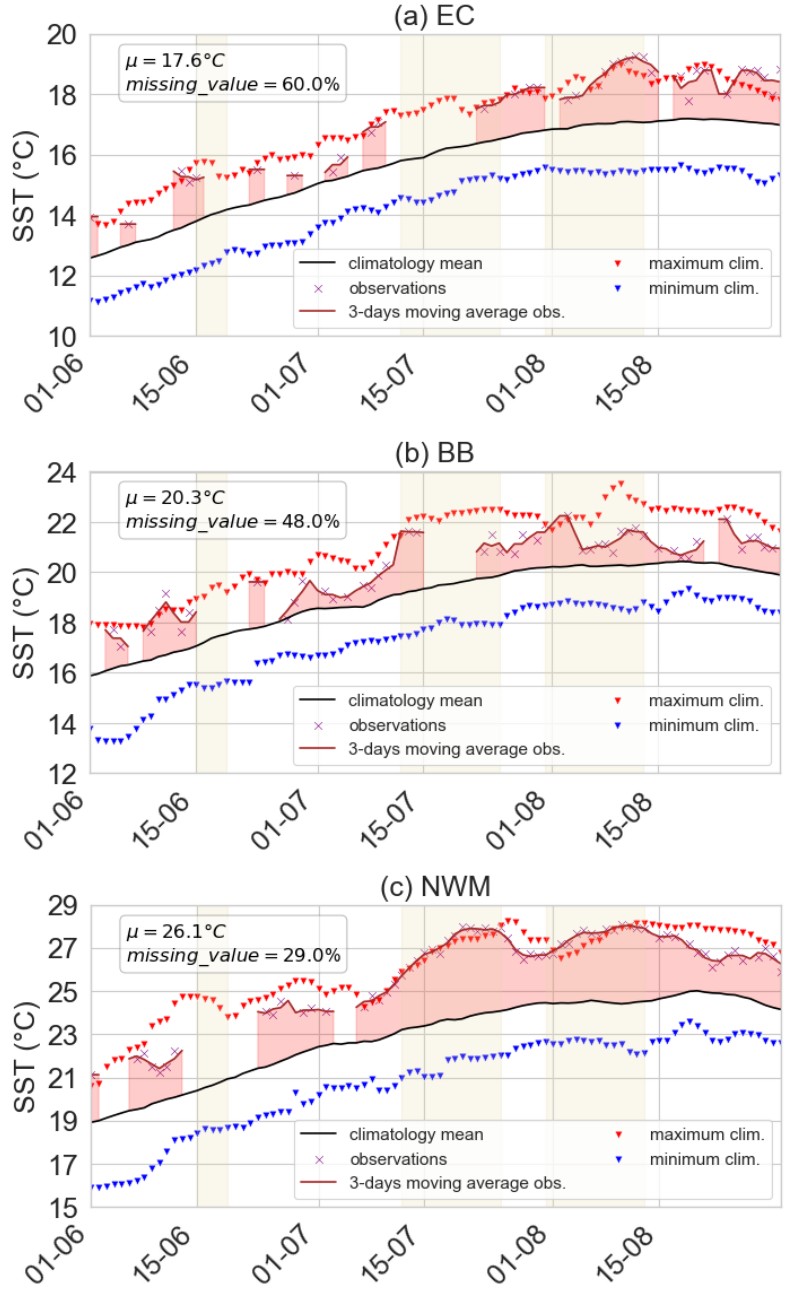

**Figure 3.** Time series of the summer 2022 daily SSTs measured by METOP-B spatially averaged over a) the English Channel (EC), b) the Bay of Biscay (BB) and c) the northwestern Mediterranean (NWM). Atmospheric heatwave periods are represented with yellow shading. Brown line is representative of the 3-day SST moving average and compared to the 1982-2011 ESA-CCI climatological mean (black line).

## 4.2 Response to the August 2022 heatwave

This section focuses on the August 2022 heatwave, as it is less affected by low data availability and all three basins can be analyzed. The heatwave started on July 31st and ended on August 13th. During this period, SSTs were abnormally high, with temperatures consistently above the climatological mean (as shown in Figure 3). Table 3 presents the spatial average SSTs and their anomalies for each basin during the August heatwave. The mean SSTs by basin (with temporal standard deviation in parentheses) over this period were 18.5°C (0.5°C), 21.5°C (0.5°C), and 27.6°C (0.4°C) for the EC, the BB and the NWM basins, respectively. The maximum spatially averaged SSTs were 19.2°C, 22.2°C, and 28.0°C, respectively. These results indicate warmer than average surface waters compared to the 1982–2011 period. In fact, the mean SST anomalies were 1.5°C (EC), 1.2°C (BB), and 3.1°C (NWM).

Positive temperature anomalies were found throughout the majority of the ocean surface and this anomaly was spatially relatively uniform (Figure4). 98% of the EC and 90% of the BB extent experienced warmer than average SSTs, while 100% of the NWM area was warmer than the climatology during this event. Few coastal areas, mainly located on the west and south coast of Brittany and continuing along the north part of the Atlantic French coasts, experienced colder than average SSTs.

We investigated the local response to the marine heatwave in each basin by calculating the 1982-2011 daily climatology for every single point within each region. Our analysis revealed that the maximum recorded temperature was 30.8°C on August 4th in the NWM area, 23.6°C on August 12th in the EC area, and 26.4°C on August 11th in the BB area. In terms of anomalies, the NWM basin exhibited the minimal anomaly of 2.2°C, whereas the EC and BB basins exhibited negative anomalies of -1.5°C and -2.1°C, respectively. The maximum anomalies were 7.9°C in NWM, 3°C in EC, and 3°C in BB, indicating the extensive response of the NWM basin and the range of sea surface temperature (SST) variability within and between each basin. The stronger SST variability in both BB and EC regions was notable.

As previously mentioned in Section 4.1, SSTs were already abnormally warm before the August 1st to August 13th heatwave. In order to identify the unique contribution of the heatwave, the weekly variation of the anomalies were computed by subtracting the anomalies of the previous week (Figure 4e and f). The anomalies variations show that the impact was pronounced in the EC, with an increase of 70% of the mean anomaly in a week (0.7°C), while the NWM area saw a weaker response of only

**Table 2.** Nighttime SST analysis over the June 1st to August 31st 2022 period. Mean values hereafter refer to spatially averaged data. Maximum represents the highest daily spatially averaged value over the 2022 meteorological summer.

| Sub-regions | SST | | | Anomaly | | | |
| --- | --- | --- | --- | --- | --- | --- | --- |
| | Mean (°C) | Max. (°C) | Std. deviation (°C) | Mean (°C) | Max. (°C) | Std. deviation (°C) | Variation coeff. |
| EC | 17.6 | 18.7 | 1.6 | 1.4 | 2.2 | 0.5 | 0.36 |
| BB | 20.3 | 21.8 | 1.1 | 1.3 | 2.4 | 0.6 | 0.46 |
| NWM | 26.1 | 28.3 | 1.9 | 2.6 | 4.3 | 0.8 | 0.31 |

15% (0.4°C). The BB basin responded firstly by a warming of the SSTs, then a sharp decline of 1.4°C between August 3rd and August 5th. The SSTs remained steady until August 9th before starting to increase again. This impacted the response of the BB, with an increase of only 20% (0.2°C) compared to the previous week

## 4.3   Observed variability of the atmospheric variables

In accordance with the previous analysis, the contribution of atmospheric variables to the persistence of abnormal sea surface temperature is analysed only during the heatwave that occurred in August 2022. As presented in the Section 2.2, the atmospheric variables are derived from the ERA5 reanalysis.

As presented in Section 3 and 2.4.1, signature of atmospheric heatwaves can be found in atmospheric variables such as the surface air temperature and the mean sea level pressure. In addition, other atmospheric variables have the potential to enhance a
315 situation that is already warmer than the climatology. As presented in the Figure 5a, we looked, in the first place, at conditions of the surface solar radiation, windspeed and total cloud cover during the heatwave that occurred in August 2022. Apart from the southeast of France, most of the area experienced positive mean surface solar radiation anomalies exceeding 50 W.m$^{-2}$ (reaching 85 W.m$^{-2}$ in southern Brittany). The daily anomaly during the period of the heatwave is significantly correlated to the anomaly of SSTs in the NWM area with a Spearman coefficient of 0.8. These spatial mean anomalies are significantly
correlated with negative total cloud cover anomalies (Figure5c, Pearson correlation coefficient of 0.90). The mean total cloud cover anomaly over France reaches -17% while the North of France and specifically Brittany have undergone the maximum average anomaly of -37%.

Regarding the mean wind speed, two zones stand out (Figure 5b). On the one hand, the BB zone experienced stronger surface wind speed on its basin (with a maximum average anomaly of 1.9 m.s$^{-1}$ during the heatwave events) compared to the August
1991-2020 average. In addition to this, the average direction of the episode was oriented in a north-easterly flow, which accentuated both the mixing and the upwelling phenomenon on the coast. This mixing accentuated the cooling of the surface waters (-1 °C between August 12th and 19th), which may explain the drop in SST in this area from August 10th to 11th (Figure3c) and the cold signal along the coast during the heatwave (Figure4).

On the other hand, the Gulf of Lions experienced a negative surface wind speed anomaly (with a minimum average anomaly

**Table 3.** Nighttime average SST analysis over the July 31th - August 13th 2022 period. Mean values hereafter refers to spatially averaged data. Maximum represents the highest daily spatially averaged value over the time period. Weekly variation corresponds to the difference between the mean anomaly calculated during the August 2022 heatwave and the one calculated during the week before

| Sub-regions | Mean SST ($\sigma$) (°C) | Mean SST anomaly (°C) | Max. anomaly (°C) | Max. SST (°C) | Weekly variation SST anomaly (°C) |
|---|---|---|---|---|---|
| EC | 18.5 (0.5) | 1.5 | 2.1 | 19.2 | 0.7 |
| BB | 21.5 (0.5) | 1.2 | 2.0 | 22.2 | 0.2 |
| NWM | 27.6 (0.4) | 3.1 | 3.6 | 28.0 | 0.4 |

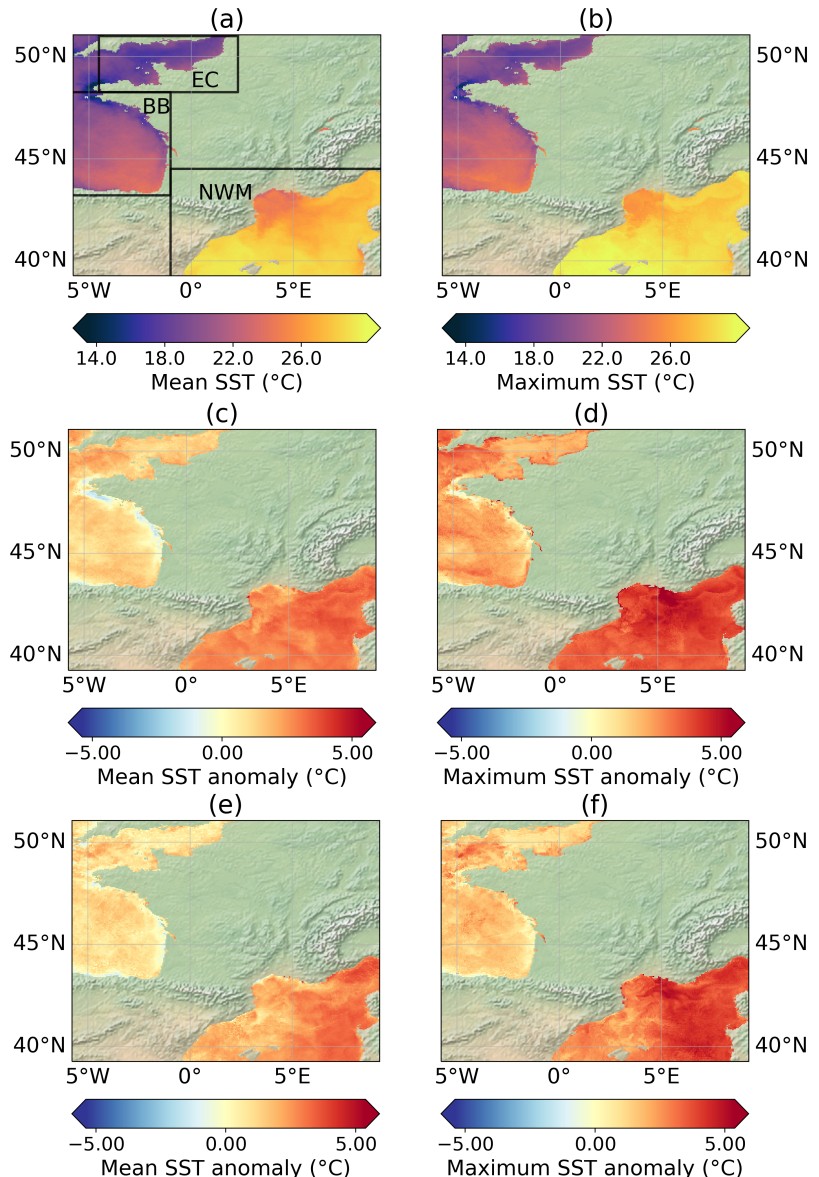

**Figure 4.** Observed SSTs and anomalies fields. (a) Mean SST values , (b) Maximum SST values, (c) Mean SST anomalies and (d) Maximum SST anomalies during the July 31th 2022 - August 13th 2022 heatwave:. (e) Mean SST anomalies and (f) during the period from July 23th to July 30th. Anomalies are relative to the corresponding 1982-2011 monthly climatology. Areas of interest used in the study. The English Channel (EC), the Bay of Biscaye (BB) and the North-western Mediterranean Sea (NWM) used in to analyse SST pattern throughout the 2022 meteorological summer are plotted on the subplot (a).

.

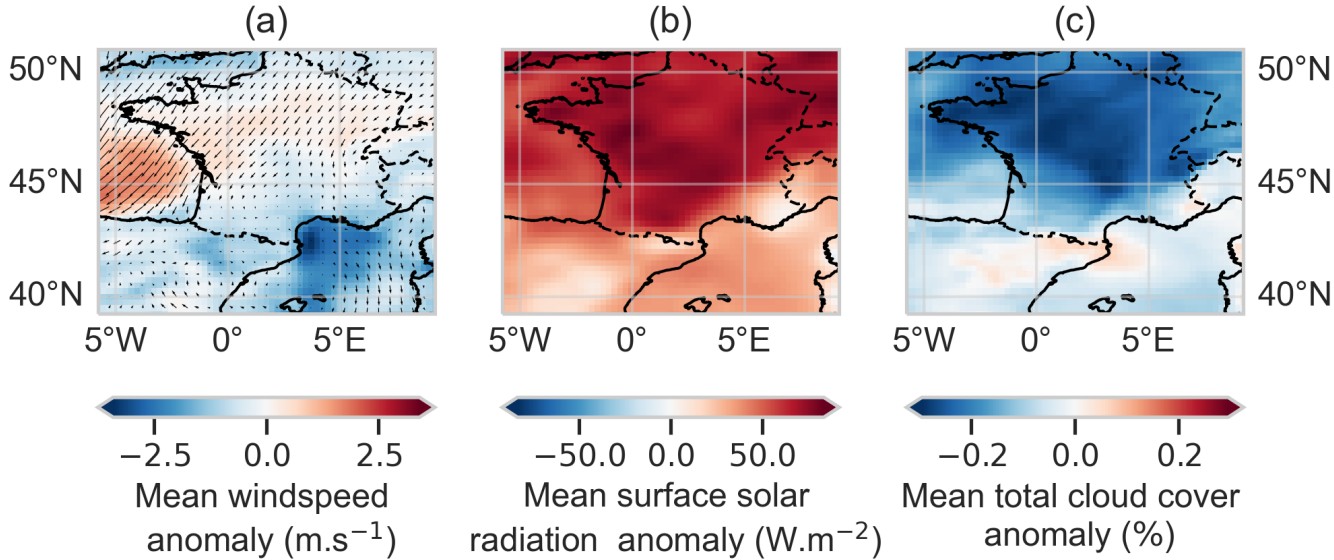

**Figure 5.** Atmospheric variables conditions during the August 2022 heatwave; a) surface solar radiation anomaly, b) 10-m wind speed anomaly and c) mean total cloud cover anomaly over France. Anomalies are compared to the same period in the 1991-2020 climatology. Data originates from the ERA5 reanalysis.

of -3.7 m.s$^{-1}$ during the heatwave events) due to a significant absence of the Mistral, oriented in a north flow, and Tramontane, oriented in a north-westerly flow, which are the main drivers of the Mediterranean turbulent mixing in the Gulf of Lions. In addition to the lack of turbulent mixing, the absence of wind could lead to a reduced contribution of turbulent latent energy flux as an ocean energy sinks. The daily anomaly over the period of the heatwave is correlated to the anomaly of SSTs in the NWM area with a Spearman coefficient of 0.6. The absence of wind could not explain in return the variability found in the Gulf of Lions. Thus this pattern can be related to the colder Rhone river outflows flowing into the Mediterranean sea.

Results for the EC are more complex to interpret because of the dynamical response to the horizontal advection initiated by the tides. From August 10th to 11th, the EC has been subject to spring tides which initiated a effective turbulent mixing, thus a decrease in the SSTs.

### 4.4 Drivers of the 2022 SSTs response

#### 4.4.1 Summer 2022 air-sea interactions

The previous section showed a significant relationship between several atmospheric variables and the SST response. In the upcoming section, the study aims at gaining a deeper understanding of the links between atmospheric variables and SSTs. The mechanisms behind the unusual warming of the SSTs are investigated using a mixed layer heat budget approach, as described in Section 2.4. The analysis is focused on the contribution of air-sea fluxes to the SST changes, with other terms deduced as a residual (see Eq.2).

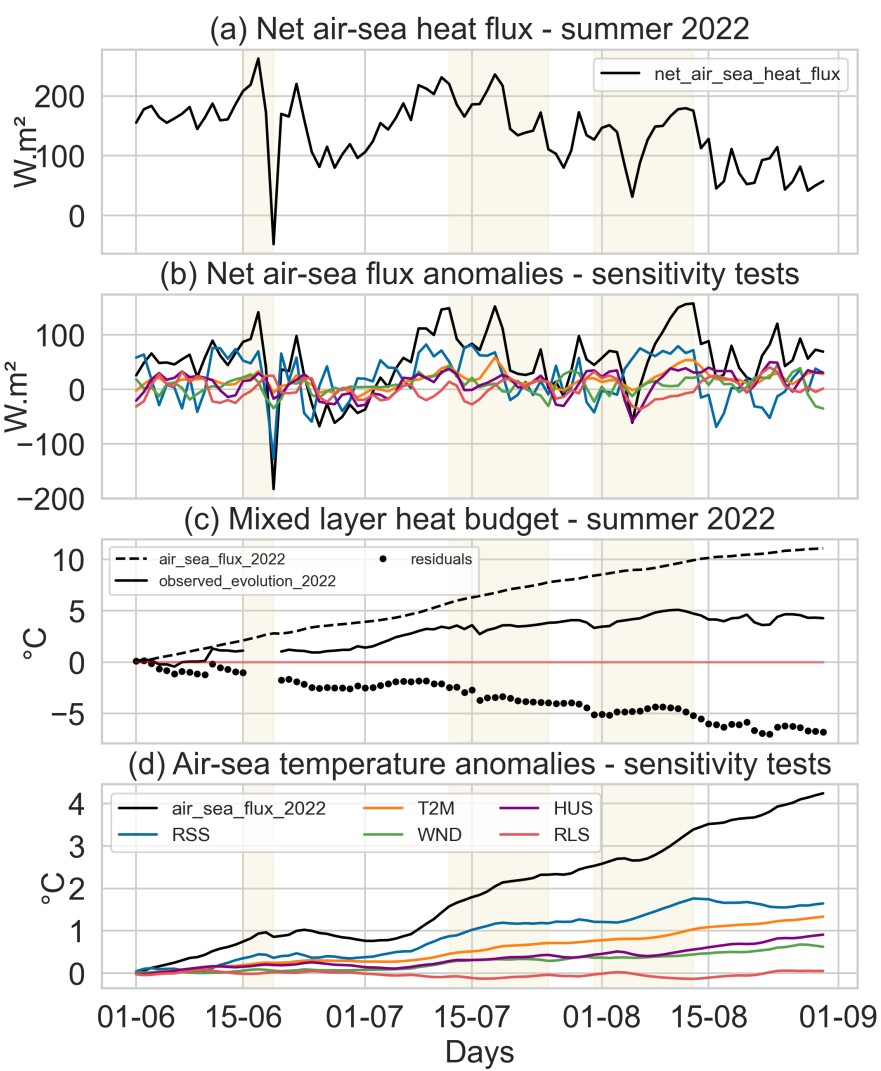

**Figure 6.** Daily EC mixed layer heat budget and air-sea flux sensitivity tests for the summer 2022 compared to the corresponding 1991-2020 climatology. (a) Simulated net air sea flux for the summer 2022 (b) Observed SST variations (solid line), contribution of air-sea fluxes to the mixed layer heat budget (dashed line) and residual (dotted line) interpreted as cooling by vertical mixing and entrainment. (b) Sensitivity test of the spatially averaged net surface heat flux anomalies compared to the CLIM experiment. (c) Same as (b) but time-integrated and expressed as an equivalent mixed layer temperature anomaly. For (b-c), RSS, T2M, WND, HUS and RLS stand for the effects of anomalous incoming solar radiation, 2-m temperature, 10m wind module 2m specific humidity and downward longwave radiation.

As illustrated in Figure 6a, Figure 7a and Figure 8a, a control experiment (CTL) was performed to evaluate the influence of air-sea fluxes on the mixed layer heat flux during the summer of 2022. The results indicate a significant contribution of atmospheric forcing to daily variability of heat flux in all three areas. Notably, the NWM exhibits the clearest response, with

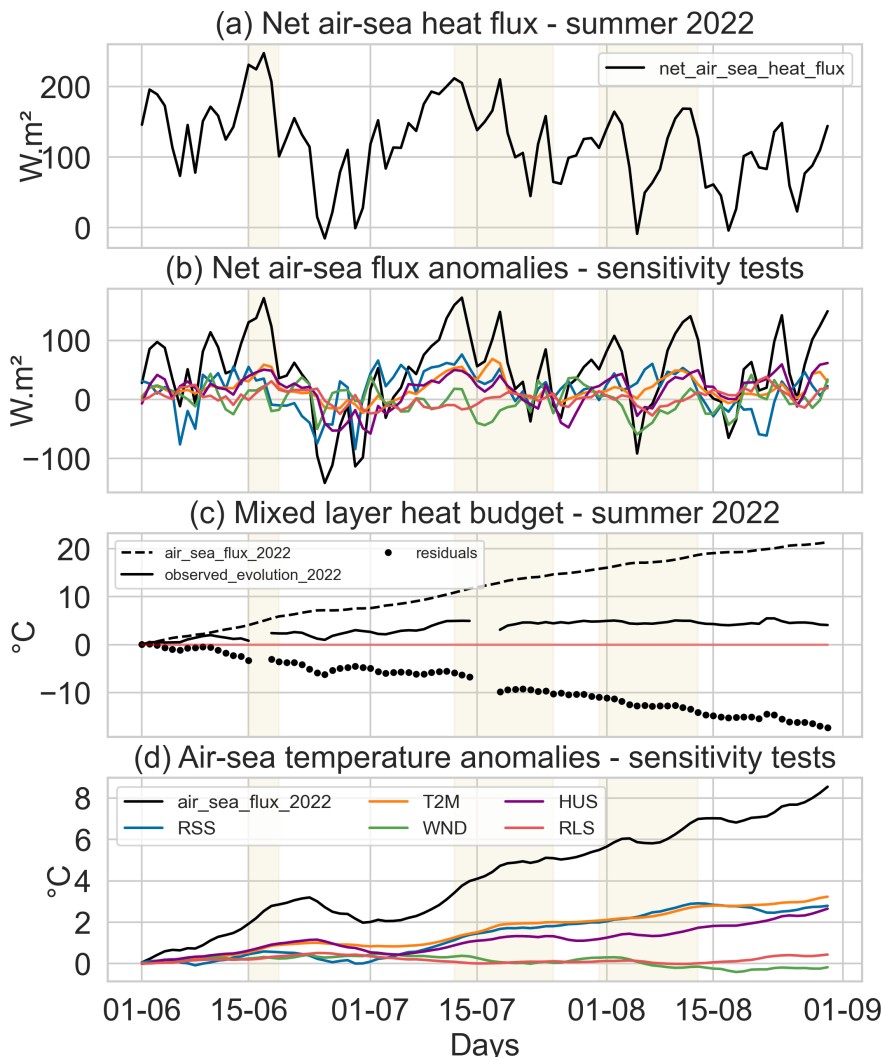

**Figure 7.** Same as the Fig 6 for the Bay of Biscay.

heatwave periods characterized by consistently high fluxes and in-between periods exhibiting stronger variability (mean standard deviation is 14.5 W.m$^{-2}$ during heatwaves whereas it becomes 49.1 W.m$^{-2}$ outside). Specifically, the mean net heat flux was 156.7 W.m$^{-2}$ during heatwaves while the mean value over the summer is 125 W.m$^{-2}$. This pattern is repeatedly observed throughout the summer, highlighting the significant contribution of atmospheric forcing. The results for EC and BB show more variability, with the net heat flux exhibiting stronger daily fluctuations. Nevertheless, the end of heatwave periods is consistently marked by a drop in air-sea fluxes, while the heatwave periods are associated with a local maximum in net fluxes.

A comparison of the net air-sea heat flux anomaly to a reconstruction based on the daily 1991-2020 climatology of surface atmospheric parameters (CLIM simulations) reveals abnormally positive values during the three heatwaves in the EC and NWM

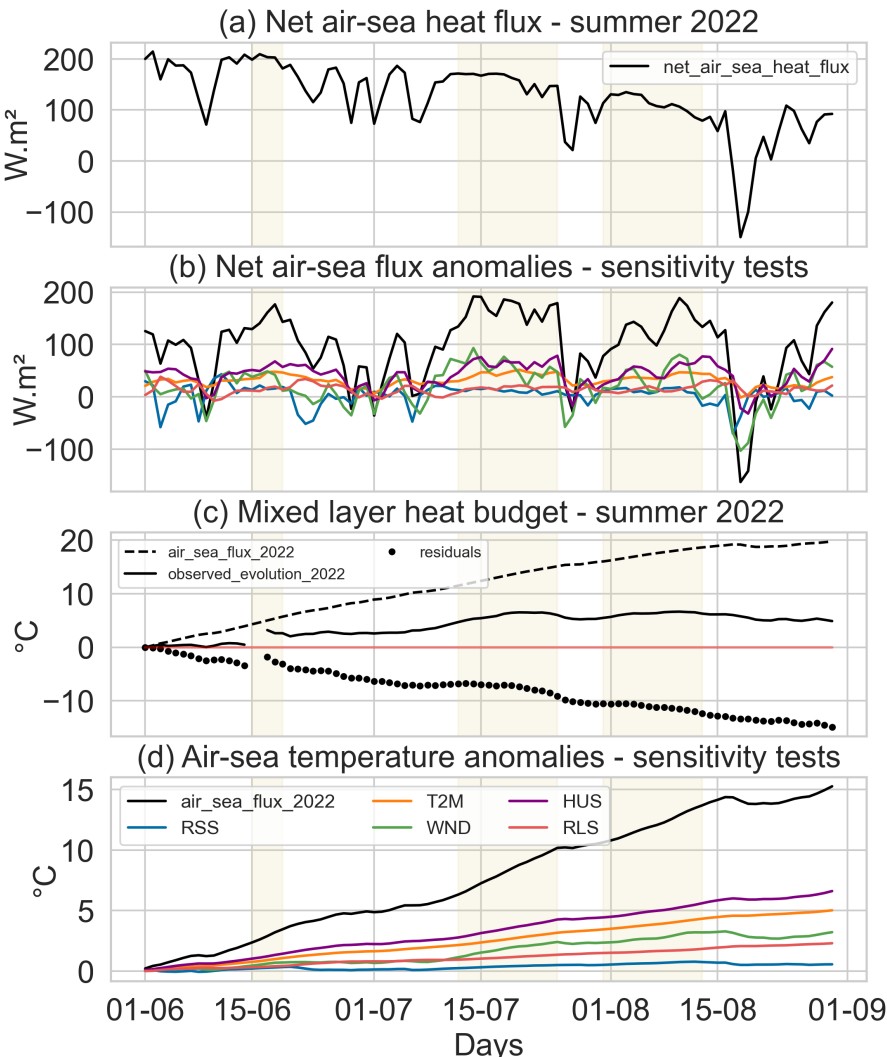

**Figure 8.** Same as the Fig 6 for the North-Western Mediterranean Sea

(Figure 6b and Figure 8b). In the BB, the stronger daily variability is associated with few days with negative anomalies during heatwaves (Figure 7b). Over the summer months of 2022, the mean anomaly was 52 W.m$^{-2}$ in EC, 47 W.m$^{-2}$ in BB and 125 W.m$^{-2}$ in NWM. The net heat flux anomaly reached its highest value during heatwave periods, with a mean of 158 W.m$^{-2}$, 86 W.m$^{-2}$, and 74 W.m$^{-2}$ over the three heatwaves in NWM, EC, and BB, respectively.

These findings underscore the crucial role of air-sea fluxes in heatwaves, and suggest that both increased atmospheric forcing and decreased cooling contributed to the anomalous mixed layer warming. Indeed, a systematic drop in the net heat budget is observed after each heatwave, with a rise before the start of the heatwave, emphasizing previous results.

Overall, the results demonstrate the significant impact of atmospheric forcing on mixed layer heat flux during the summer of

2022, and provide insight into the complex dynamics of air-sea interactions during heatwaves.

This dynamic of the atmospheric fluxes has an impact on the average mixed layer temperature evolution (term E in Eq.1) as illustrated in the Figure 6c, Figure 7c and Figure 8c. In each of the areas, the air-sea flux-induced mixed layer temperature trend exhibited a continuous warming trend, resulting in a tendency of 0.12 $°C.day^{-1}$ for EC, 0.23 $°C.day^{-1}$ for BB and
0.16$°C.day^{-1}$ for NWM over the summer period. By comparison, the observed SST trend (left-hand side term in Eq.1) was estimated to be 0.07 $°C.day^{-1}$ for EC, 0.04 $°C.day^{-1}$ for BB and 0.08$°C.day^{-1}$ for NWM. The difference between these two terms represents the residual term, "Res", in Eq.1. Here this term is negative, indicating that ocean processes tend to cool the upper ocean. The mixed layer temperatures in the EC and BB areas were strongly influenced by residual term (the sum of all dynamic and diffusive terms), with a stronger contribution over longer periods, likely due to stronger cooling processes such
as vertical mixing. The trends observed during the heatwaves provide valuable insights into the complex dynamics of different regions. For example, between July 12th and July 25th (July heatwave), the mixed layer temperature trend was weakly positive for EC (0.04 $°C.day^{-1}$) and even negative for BB (-0.01 $°C.day^{-1}$). In contrast, during heatwave periods over NWM, the residual term remained relatively constant, while the mixed layer temperature trend slightly increased, demonstrating the significant role of surface fluxes in driving the warming. For instance, in the NWM, between July 12th and July 25th (July
heatwave), the air-sea flux contribution reached 0.18$°C.day^{-1}$, and between July 31th and August 13th (August heatwave), it was 0.19$°C.day^{-1}$, compared to a trend of only 0.07$°C.day^{-1}$ between these two heatwaves. Thereby the air-sea fluxes implied a clear warming trend suggesting the non-negligible contribution of ocean cooling processes to compensate for the atmospheric forcing and preventing SSTs from reaching record-breaking levels for longer periods. Almost invariably, the end of heatwave periods is associated with a cooling residual, suggesting a termination of events through vertical oceanic mixing
while the air-sea flux budget becomes less effective.

### 4.4.2    Disentangling atmospheric variables contributions

To gain a deeper understanding of the processes contributing to the mixed layer temperature increase, we examined the contribution of atmospheric fields to the daily net air-sea heat flux (illustrated in Figure 6b, Figure 7b and Figure 8b and the associated
air-sea mixed layer temperature trend (illustrated in Figure 6d, Figure 7d and Figure 8d in comparison to a reconstruction based on the daily 1991-2020 climatology of surface atmospheric parameters.
The sensitivity test results, displayed in Figures 6(c,d) and 7(c,d) reveals that the anomalous net air-sea flux in the EC and BB can be attributed mainly to the near-surface air temperature, the surface solar radiation and the near-surface specific humidity. In both basins, the radiative budget dominates the heat budget along with the contribution from surface air temperature. Specif-
ically, in EC, SSR accounts for 39% of the total mixed layer temperature trend anomaly, while surface air temperature and specific humidity contribute 31% and 21%, respectively. In BB, surface air temperature is the largest contributor at 38%, followed closely by RSS at 35%, and specific humidity at 31%. These findings highlight the critical role of the radiative fluxes in shaping the heat budget and temperature trends in these regions. Our observations align with the positive anomalies in surface

solar radiations recorded in both areas (Figure 5b). The contribution of specific humidity in the BB heat budget is explained by the moistening of the air mass during the advection of hot air masses from the northeast. Wind speed also plays a similar role in decreasing the net air-sea flux at the end of each heatwave in both EC and BB. However, the wind speed contribution to EC net air-sea flux is relatively low in the termination of heatwaves. In contrast with EC, wind speed is a key factor in limiting the response of the SSTs in BB during heatwaves. For instance, the drop in the net air-sea flux anomalies during the heatwave of August is mainly attributed to an anomaly of wind speed that enhanced the turbulent fluxes and thus the heat loss. This results confirmed the effect of the positive wind speed anomaly in BB (Figure 5a) which increase the coastal upwellings and reduce the specific humidity contribution (north-easterly advection of continental air masses) As an example, both June and August heatwave endings are attributed to the negative anomaly of the surface solar radiation in EC.

The sensitivity test results, displayed in Figures 8b and 8d indicate that the anomalous net air-sea flux in the NWM can be attributed mainly to the near-surface specific humidity, the near-surface air temperature and the near-surface wind speed anomalies. The specific humidity accounts for 43%, surface air temperature explains 33%, and wind accounts for 21% of the total mixed layer temperature trend anomaly. These three atmospheric variables have a significant impact on turbulent fluxes, with the latent heat fluxes showing a stronger response (Figure A3). The effect of wind speed (WND) is twofold, it impacts the atmospheric heat fluxes but it also impacts the transfer of momentum to the ocean and therefore the ocean processes. Over the NWM in August, wind speed is weaker than normal and drives a reduced latent heat flux which tends to warm the ocean. We have not estimated its effect on momentum and ocean dynamics but a reduced wind speed in the region is most probably associated to reduced ocean surface cooling by mixing. This is consistent with Figure 5 that shows a negative correlation between wind speed and SST. Conversely, wind speed was the main contributor (reaching a contribution of 63% on August 18th) to the drop in the net air-sea flux at the end of each heatwave, while 2-m air temperature (T2M) and 2-m specific humidity (HUS) have little effect. Wind speed may also contribute to the negative heat trend seen in the residual term at the end of heatwaves, likely linked to wind-driven turbulent vertical mixing and entrainment. Surface air temperature and specific humidity are closely linked to the synoptic conditions in this specific regions. As detailed in Section 3, the observed increase in specific humidity can be attributed to the moistening of the air mass linked to its southerly advection over a maritime region. Upon reaching the study area, the air mass was abnormally hot and humid, which in turn contributed to the air-sea flux forcing.

The results demonstrate that while the record-breaking surface air temperature has a major impact on the area, the contributions of surface humidity and wind speed should not be underestimated. Contrary to the EC and BB areas, the contribution of the surface solar radiation is relatively low but to the overall net air-sea flux is consistent with the non significant anomaly of RSS observed over the area in August (Figure 5b). Thereby, the surface solar radiation did not act as a key contributor to the SSTs response in NWM during heatwaves.

## 5 Discussion & Conclusions

Using OSI SAF satellite products and an ocean mixed layer bulk model, we investigated the response of SSTs to the heatwaves that occurred during the summer of 2022 and the extent to which they could be attributed to changes in atmospheric variables. This is the first study to provide insights into the ocean thermal response to the exceptional summer of 2022 at a country scale, integrating sub-areas with distinct characteristics, including the English Channel (EC), Bay of Biscay (BB), and Northwestern Mediterranean (NWM) Sea.

Despite the significant lack of data, particularly in the early summer and in the EC area, we found a clear warming signal of SSTs during the summer of 2022 that was evident in all studied areas. All three areas exhibited positive SST anomalies throughout the summer, with record-breaking daily anomalies indicating that 2022 was one of the warmest summers in terms of SSTs, which also started early in the season. The strongest response was found in the NWM, with a seasonal SST anomaly of 2.6 °C, reaching 3.9 °C during the heatwave that occurred in August and exceeding the climatological maximum for 22 summer days. It should be noted that the climatology takes into account summer 2003, which may be a record for SSTs in the NWM. Locally, these SSTs were even higher, with a peak measured at 30.8 °C. A similar pattern is found for the EC area, which was consistently close to the daily climatological records, reaching a maximum anomaly of 2.2 °C in August and 19 days over the climatological maximum, with a mean summer anomaly of 1.5 °C. The response of the BB area is lower in magnitude, even if the mean SST anomaly is 1.2° °C and reaches a peak of 2.4 °C, with only 4 days above the climatological maximum. This study demonstrates that the summer of 2022 was one of the warmest summers in terms of SSTs. In the specific case of NWM, this mark a record over the period 1982-2011 with a mean temperature of 26.1 °C. The response of SSTs in the Mediterranean Sea has been extensively studied and our results over this area are in line with previous studies investigating the contribution of heatwaves to Mediterranean SSTs, such as the 2003 heatwave studied by Olita et al. (2007). Focusing on the Central Mediterranean Sea, they found similar magnitudes in the SST warming with mean anomalies around 2°C. The Mediterranean Sea is recognized as a "hotspot" for climate change (Giorgi, 2006), which will face warmer summer seasons (Adloff et al., 2015). Our findings support the idea that the occurrence of heatwaves throughout the summer would cause the NWM Sea to respond strongly to these atmospheric forcings. Indeed, results indicate that even during non-heatwave periods, the SSTs in the NWM area were consistently warmer than the climatological average, even when the net heat flux was close to normal. These results are also in line with the with observations (Bensoussan et al., 2019) and modeled evolution (Darmaraki et al., 2019) of the continuous warming of the Mediterranean Sea. Regarding EC and BB, this study contributes to the limited body of research on the responses of these areas to external factors. Our results align with previous studies that have emphasized the significant role of regional hydrodynamics in shaping SSTs (Izquierdo et al., 2022)

The warming of upper layer ocean temperatures during summer is mainly driven by the occurrence of atmospheric blocking related to the multi-decadal variability of the North Atlantic (Häkkinen et al., 2011). This implies that atmospheric forcings are a key contributor to the mixed layer heat budget in most ocean areas (Salinger et al., 2019; Amaya et al., 2020). Such events, which are characterized by an atmospheric blocking system associated with above-climatology surface air temperature and low surface wind speed, result in a warming of SSTs between 2°C and 4°C, that is the range we found for the SSTs anomalies

during the summer of 2022. The associated fluxes are key indicators of the ocean's rapid response to extreme atmospheric conditions. Our findings suggest that the response of SSTs varies significantly between different ocean basins and is closely linked to the specific environmental characteristics of each region. While all areas respond strongly to record-breaking temperatures, the advection of hot and humid air masses is the primary driver of SSTs in the NWM area, whereas areas such as EC and BB are more reactive to the variability of surface solar radiation. In turn, the role of wind on air-sea heat fluxes is variable between basins and generally second-order. The increase of SSTs over NWM during the summer of 2022 can be attributed mainly to an increase in surface air temperature and specific humidity, and a decrease in wind speed, which reduced the effectiveness of turbulent heat fluxes as an ocean heat sink. Compared to a control experiment where all atmospheric variables are set to their 1991-2020 climatological values, the (positive) radiative heat fluxes increased slightly over the course of the summer, while the (negative) turbulent heat fluxes reached minimum values during heatwave periods. For the EC and BB areas, the contribution to the warming of SSTs is dominated by surface air temperature and surface solar radiation, which in turn confer a stronger role to the radiative fluxes in the mixed layer heat budget. This is explained by the greater interannual variability of solar radiation in summer over these basins, whereas for NWM, radiation is close to its maximum value every summer. It should be noted that the contribution of specific humidity is not negligible and can be explained by the wind direction, which initiated moistening of the air masses over these areas. As shown for the NWM area, the patterns of the different flux terms in BB and EC are similar, with negative anomalies of turbulent fluxes and positive anomalies of radiative fluxes during heatwave periods, while the opposite occurs in between. The observed pattern triggers a marked increase in positive anomalies of the MLD heat budget during heatwaves. Notably, our findings reveal that the MLD heat budget is unusually high in cases where turbulent heat fluxes fail to act as heat sinks, while radiative fluxes enhance atmospheric forcings. In addition, the pattern of SSTs response is modulated by the residual terms which are reduced by atmospheric forcing during heatwaves. These terms are linked to oceanic cooling mechanisms such as vertical mixing. The observed trend of temperature extreme events highlights the crucial role of air-sea fluxes in the formation of record-breaking SSTs, with residual terms emerging as the dominant contributors by the end of each heatwave period. These findings indicate that rapid warming of SSTs occurred during the summer months due to an imbalance in surface heat fluxes, which were intensified by specific atmospheric heatwave conditions and modulated by regional processes accounted for in the residual terms.

Our results are consistent with the atmospheric circulation patterns observed during heatwaves. During the summer of 2022, atmospheric blocking occurred simultaneously with a southerly advection initiated by cold drops off the Iberian Peninsula, resulting from a static meander of the jet stream, which enhanced the heatwaves that contributed to record-breaking SSTs. These high-pressure systems also have regional-scale impacts, such as modulating surface wind speed. Even though wind speed is not the main driver in all the areas in our study, negative anomalies in wind speed (and corresponding latent heat negative anomalies) are correlated with the initiation, while positive anomalies are correlated with the cessation of marine heatwaves (Sen Gupta et al., 2020). In the presence of negative wind anomalies, the turbulent mixing of surface waters becomes less efficient and could result in thermal stratification that would limit exchanges with colder subsurface waters. This effect might be particularly pronounced in areas such as the Mediterranean, where tides are almost absent, as opposed to the EC, where

tides influence SSTs, as demonstrated at the Ushant Front (Chevallier et al., 2014; Karagiorgos et al., 2020). The pattern of the radiative fluxes can be explained by the atmospheric subsidence, which tends to lower relative humidity, reduce total cloud cover, and warm the surface air temperature. Besides the subsidence, the southerly advection, linked to the Iberian cut-off low, over the warm Mediterranean Sea enhanced the moistening of the air mass and increase the specific humidity thus the turbulent heat fluxes.

The magnitude of anomalies can also be attributed to anthropogenic forcing, which can be quantified using singular event detection and attribution cutting-edge approaches (Ribes et al., 2020; Faranda et al., 2022).

The approach proposed in this study is sensitive to missing data caused by cloud cover or atmospheric aerosols. To anticipate the detection of anomalously warm SSTs, we could have used combined products such as Operational Sea Surface Temperature and Sea Ice Analysis (OSTIA) (Donlon et al., 2012). However, interpolated gap-free global SST data products might end up hiding specific trends or introducing biases in the SST analysis (Stobart et al., 2015). A possible improvement could be to combine satellite observations with a specific model, such as the one developed by Hobday et al. (2016), to detect large-scale anomalously warm SSTs. Currently, the detection model is unidimensional, and development of a simple dynamical two-dimensional model would help track water temperature anomalies in the upper-ocean layer and understand temperature feedback within the mixed layer. More generally, development activities must continue with the implementation of more advanced models capable of assimilating these observations and incorporating physical and biogeochemical processes, which will increase the accuracy and reliability of monitoring SSTs. This will provide a more general view of the processes and feedbacks involved, but also access to the responses of the surface layer of the ocean where there is a systematic gap in observations. Daily monitoring of SSTs will be crucial for understanding and forecasting changes in biological responses at regional and global scales (Doney et al., 2012), including the detection of harmful algal blooms that might endanger several coastal zones, shifts in community organization, and prevention of mass mortalities of endemic species (Garrabou et al., 2022; Smith et al., 2022).

Improving observation techniques will also play a role in enhancing SST monitoring. In this study, only data from the polar-orbiting satellite Metop-B were used. Synergies between polar-orbiting and geostationary satellites will provide a significant gain for SST monitoring (Vanhellemont et al., 2014; Minnett et al., 2019), as polar-orbiting satellites have high spatial resolution but limited temporal resolution, while geostationary satellites have high temporal resolution but limited spatial resolution. The new generation of meteorological satellites, such as Meteosat Third Generation and Metop-SG, will offer improved monitoring in both time and space with interesting nominal resolutions to locate systems with smaller characteristic lengths while being able to monitor larger scale systems with ideal temporal resolutions for operational purposes (Holmlund et al., 2021). This highlights the need to work on operational SST products, such as those developed within the OSI SAF project, and to contribute to new products, such as ocean color, in the fields of biogeochemistry and physics, to prevent deep alterations to the oceans in the context of climate change.

Furthermore, in this study, we strictly analysed the atmospheric contribution. Here the aim of the study was to disentangle the respective roles of surface atmospheric variables in explaining the SST anomalies in a forced ocean context. However, we can speculate on the link between forcing and dynamic ocean processes. For example, the net heat budget variability is driven by wind speed outside of heatwave periods associated with a decrease in SST anomalies. Therefore, we can hypothesise about its impact on cooling terms (referred to as residuals in the study) and the non-negligible role of other drivers, including turbulent vertical mixing and entrainment, vertical and horizontal advection by regional current systems, in the upper-ocean heat budget. Nonetheless, we did not assess how these atmospheric forcings could propagate inside the ocean and what would be the feedback from SSTs themselves. The mixed layer depth used has a lower bound of 10m, which might conceal important features that could explain the higher warming trend in mixed layer temperature in response to short-term atmospheric forcing variability (Amaya et al., 2020). For this purpose, we would need to have access to long-term measurement means capable of capturing processes occurring below the surface in addition to complex unidimensional and 3D ocean models capable of representing turbulent mixing, entrainment, and complex features such as upwellings. The study did not quantify this impact and further development would be necessary to assess how much the turbulent and the advective oceanic heat fluxes have been lowered during the summer of 2022. Although a comprehensive understanding of the mechanism between the atmosphere and the ocean can be gained with a coupled model analysis. Hence, a comprehensive model analysis constrained by subsurface observations is needed to integrate the various drivers influencing the upper-ocean heat budget.

This study provides insights into the response of SSTs during the 2022 meteorological summer and the link with synoptic conditions in France. However, it should be acknowledged that the abnormal meteorological situation persisted throughout most of the autumn season in France, with surface air temperatures continuously above long-term averages. Furthermore, a systematic study of the different areas, including additional months, could help assess the implications of persistent atmospheric heatwaves and their regional dependence.

*Code and data availability.* All post-processing codes are available on Zenodo in the following repository: https://10.5281/zenodo.7194099 (Guinaldo, 2022).

Data used in this study are open-source and freely available. OSI SAF L3b can be found here: https://osi-saf.eumetsat.int/products/osi-201-b. ERA5 reanalysis can be found here: https://cds.climate.copernicus.eu/. ESA-CCI SST product can be found here: https://catalogue.ceda.ac.uk/uuid/62c0f97b1eac4e0197a674870afe1ee6. All these data were accessible on the date of the manuscript submission.

# Appendix A: Supplementary figures

## A1 Heat budget terms sensitivity tests

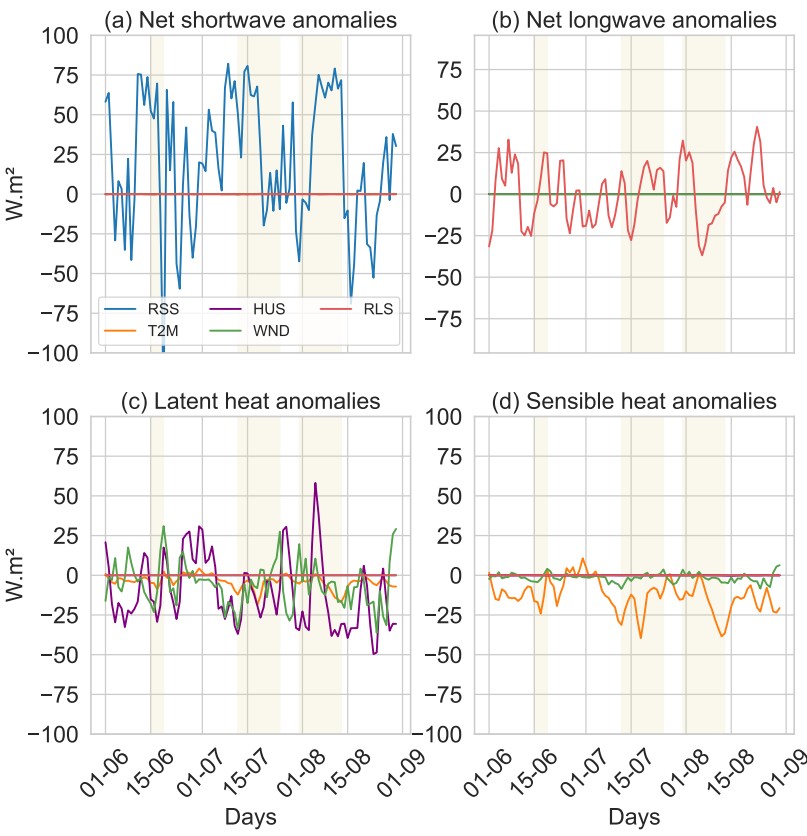

**Figure A1.** Decomposition of the net air-sea heat budget over the EC area during the summer 2022 for all sensitivity experiments compared to the CLIM experiment.

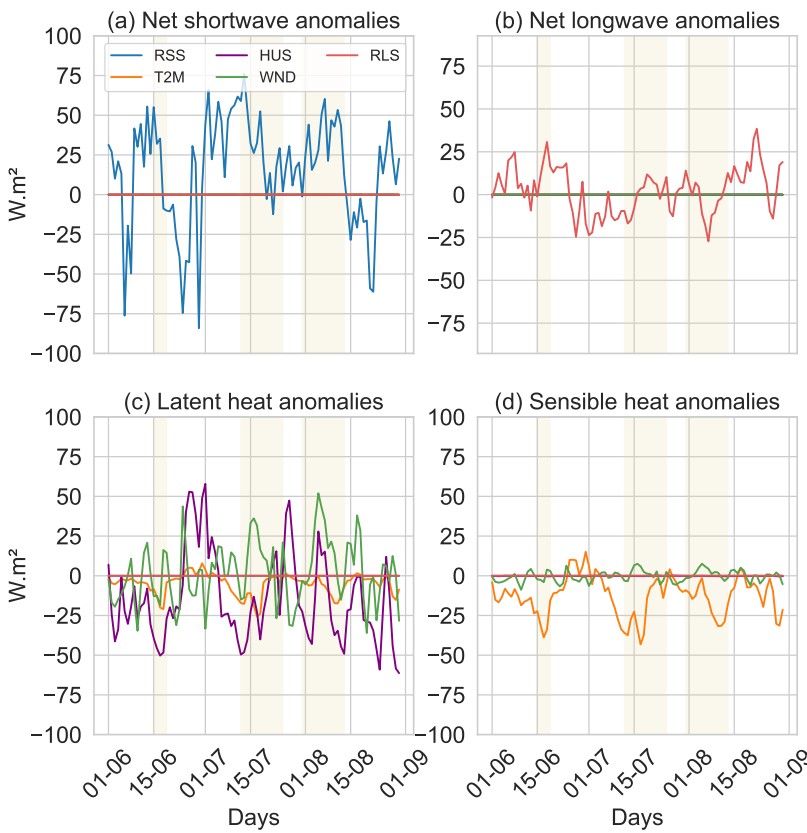

**Figure A2.** Decomposition of the net air-sea heat budget over the BB area during the summer 2022 for all sensitivity experiments compared to the CLIM experiment.

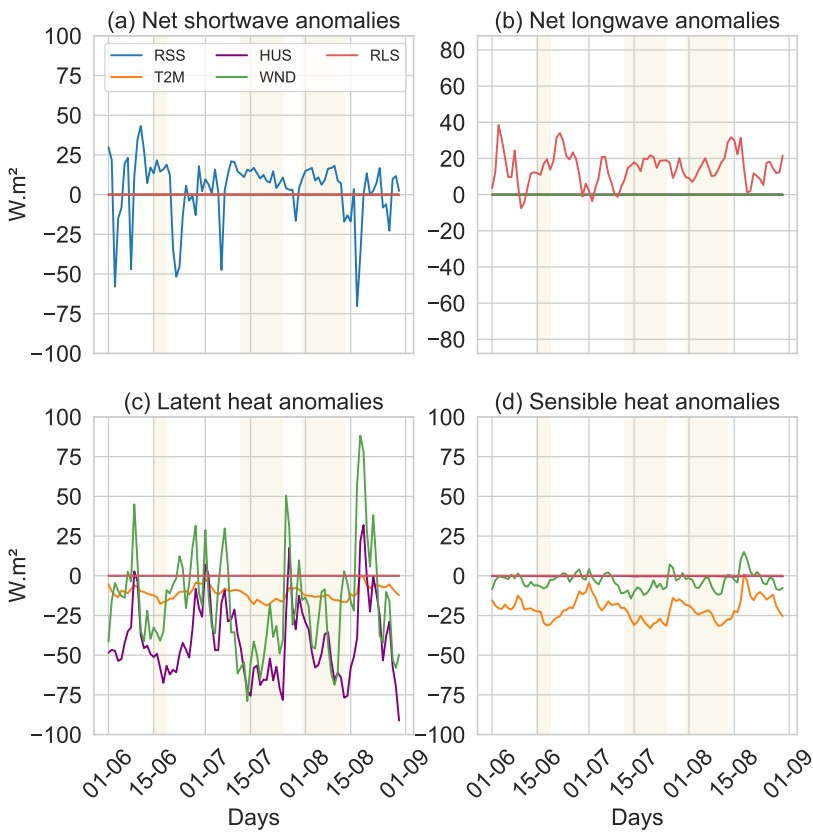

**Figure A3.** Decomposition of the net air-sea heat budget over the NWM area during the summer 2022 for all sensitivity experiments compared to the CLIM experiment.

## A2 Mixed layer depth variability during the summer of 2022

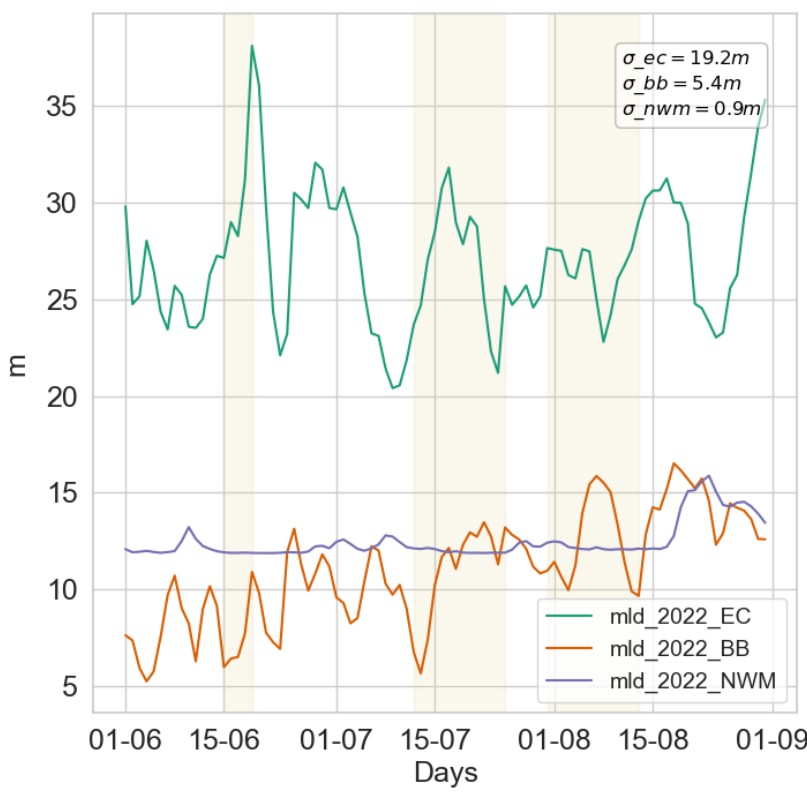

**Figure A4.** Daily mixed layer depth analysis from the CMEMS Mediterranean Forecasting System (Med-Physics).

*Author contributions.* TG, SS and HR designed the satellite observation study, TG carried the observation analysis out. TG, AV and RB carried the modeling part out. AV produced the forcings dataset and carried out the numerical simulations. TG developed the code and performed the analysis of both observations and modelling outputs. TG wrote the manuscript with contributions from SS, AV and RB. All co-authors took part on discussions and revisions of the manuscript.

*Competing interests.* The authors declare that they have no conflict of interest

*Acknowledgements.* The authors express their gratitude for the financial support provided by EUMETSAT through the Ocean and Sea Ice Satellite Applications Facilities (OSI SAF), which have been instrumental in the development of operational sea surface temperature products. Additionally, TG extends special thanks to Samuel Somot for his valuable time and insights in discussing marine heatwaves.

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
