# Peer review of "Response of the sea surface temperature to heatwaves during the France 2022 meteorological summer"

_EGUsphere, 2022_

## Referee Comment (RC1)

**Review of "Response of the sea surface temperature to heatwaves during the France 2022 meteorological summer"**

**General comments**

In this manuscript the authors address an interesting, and not extensively studied, question, the response of SST to atmospheric forcings under atmospheric heatwaves. One of the reasons why this work is of interest is its immediacy, as it deals with the situation in the summer of 2022. However, this may also be a weakness as it uses data from near-real-time operational systems that may suffer from some validation or availability problems.

The authors state that their intention is "*to determine the ability of satellite measurements 5 to track the response of surface waters to these events*" right from the abstract but, across the manuscript I can't see if such ability has been validated or if this improves the results that could be achieved using historical (non-operational data). And more importantly, I can't see the need to track the SST response to atmospheric forcing in an operational environment as SST itself is routinely measured and/or modelled.

Two different SST data sets are used, to my understanding, for building SST climatology and as operational SST. If they are to be compared, some explanations are needed in the text for the need to use different data sets and how they are correlated/validated against each other. I know about these products and absolutely rely on their quality, just asking to include some more information on the data and methodology used.

I find that the authors have done an interesting job that is worth to be published with major revision. I am especially concerned about the final objective, or question to be answered, in the manuscript. I could not find if the most important results come from the analysis of SST response to atmospheric forcings (which I find quite interesting results) or from the "ability" of operational SST data to track such forcings. If you think the operational interest of such tracking is important, please try to reinforce your findings and justify the need for this tracking, I don't clearly see the interest but it is just my opinion.

Please, try to get more focus in the points you are interested in and fully rethink the manuscript structure and final objective. There is a lot of work beneath that would be good for publishing but maybe the paper does not actually reflect that but a mix of questions/objectives so the reader does not get a clear conclusion.

Looking forward to receive the revised manuscript.

Please, think about paper structure and writing. I found parts of the discussion that would fit better in the introduction section. See comments below. I don't like to find so many references in the conclusions. Conclusion section is just for you, to explain your findings.

**Minor comments**

Please, carefully check the English grammar and spelling. And then check again.

Across the manuscript, please clearly state how do you calculate anomalies. Are you comparing daily OSI/SAF SST values with monthly climatology? I couldn't get the point.

Figures are too small. Please, make bigger figures.

**Other comments**

Lines 45-54 should not be included in the Introduction Section. A "Synoptic description/atmospheric characterization/…" section is needed and should go into greater detail.

Figures 1-2: I assume they are monthly anomalies although not stated. Maps are too small.

Figure 3: Please, improve caption by stating the subdivisions are stated for the SST analysis

Lines 69-71. Please, rewrite this sentence to improve the English language (missing verbs, concordance,…).

Lines 76-77: Change "Both hourly and monthly were used" to "Both hourly and monthly **data/values** were used" (general remark: please, carefully check English grammar and spelling)

Line 142: Please change "3°CC"

Line 153: Reference to "Table 4" but this one does not exist. Maybe A1?

Lines 160-164: Do these sentences refer to single point values? The 7.9ºC anomaly, is referred to local or mean climatology? This is not clear for me. If this is compared to mean areal climatology this would not be the right way to compare daily values to climatology.

Line 179: Which is the "whole domain"?

Figure 7: Not stated but I understand that the maps show anomalies for the days in August 22 heatwave days, respect to 1991-2020 august monthly values? To the same period (31/07-13/08) for the 1991-2020? Please add this information in the figure caption and text

Line 209: *in response to the atmospheric heatwaves that affected France during the 2022 summer.* Results and analysis are mostly centred in the August event, maybe not valid for the rest of summer

Lines 209-213: *"The strongest response was found 210 on the NWM basin (with a maximum average SST anomaly of 4.3°C) which is in line with observations (Bensoussan et al., 2019) and modeled evolution (Darmaraki et al., 2019b) confirming the Mediterranean Sea is a "hotspot" for climate change (Giorgi, 2006)"*. A single extreme event does not confirm the hotspot, although I agree with the sense of your assumption and that it is in line with the cited references. Please, rewrite this sentence.

Lines 227-244: These lines are not a discussion/conclusion but should be part of the introduction section. No work has been done regarding MHWs across the manuscript.

Line 246 *"To prevent the detection of anomalously warm SSTs"*. To prevent?

Lines 250-256: Rephrase to improve reader understanding

---

## Referee Comment (RC2)

**Object**: Referee report for Ocean Science Copernicus publications

**Paper**: Response of the sea surface temperature to heatwaves during the France 2022 meteorological summer

**Authors**: Thibault Guinaldo, Stéphane Saux Picart, and Hervé Roquet
* * *
General comments

The authors present a study of SST response to anomalously warm meteorological summer of 2022 in France. In particular, the SST response study is performed in three regions, the English Channel, the Bay of Biscale and the North Western Mediterranean Sea, by analysing NRT SST data compared to historical climatology based on L4 ESA CCI SST product. Also, they analyse additional essential variables to complete the whole area overview during summer 2022, by considering Surface solar radiation, windspeed and cloud cover.

The topic addressed is of importance and interest, given the known urgency to understand marine extreme events, to deepen the comprehension of atmosphere and ocean interactions during such events, together with the fact that summer 2022 has been showing record-breaking temperatures both on land and on sea, and is just at present starting to get back to climatological values.

In my opinion the work presented does have interesting outcomes and comments, being also one of the first works which addresses the extreme event of summer 2022, but at the present version lacks in defining clearly its main objective, also presenting results in such a way which makes it difficult to follow smoothly. I should suggest a major revision prior to publication, finding that improvements in presentation have to be considered to make the manuscript more robust, hence more impactful. I hereafter give major comments, followed by line-by-line suggestions to ease revision.

- You talk about summer 2022 heat wave, some times as a whole event, some other times as several separated events with specific dates, and some other times as separate events for each month of June, July and August. I would suggest that you try clarifying how you separate or put together the anomalous event. This would be very useful for the reader's overall understanding.
- The overall objective of your work is not completely clear: Are you willing to demonstrate the need of an operational NRT SST product, or are you mainly interested in investigating the role of ocean response to atmospheric extreme event? This is not clear since your presentation seems to address a bit of both but not really focusing or one or the other.
- Some of the numeric results presented need to be additionally explained or properly references to Figures and Tables to aid readability and understanding (see line by line comments)
- A long part of Discussion and Conclusion section is more properly read as introductive. I suggest to carefully revise this section.
- English must be carefully revised, for typos and full sentences meaning. I report some corrections in line by line comments, but they are not exhaustive.
* * *
Line by line comments

Abstract

(1-3) These initial sentences should be more precise. From the beginning of which measurements? The Seasonal average of 22.7°C is intended as surface air temperature, or what? Multiple record-breaking heat waves over which period?

(9-10) I would not introduce the core of your study saying that "the contribution of other atmospheric variables is not negligible". This sounds like an a-priori consideration. I believe, instead, that the choices of variables to investigate have a clear motivation and should be better introduced and made more attractive in the Abstract.

Introduction

(Figures 1 & 2) The images should be larger, could aid having titles referring to the months. How did you compute the 1991-2020 climatology to which you define anomalies? Since the main rationale of your work in exactly the anomaly in surface air temperature, consider if to switch order of Figures 1 & 2, or if to add a single Figure before these showing the mean spatial anomaly over whole summer, or if a time series.

Study sites

(25-32) You longly talk about OHC and the reader it brought to think he/she will see some analysis on OHC, which instead is not present. I suggest you shorten OHC description, or you add an analysis of the field in summer 2022

(45-48) As presented this part sounds as if you are describing already some results, because you go specific on dates and causality of events with no reference. I suggest you keep introduction to be more descriptive, eventually recalling in results more specific relations of causation

(64) Since you introduce the SST acronym at line 37, you should not repeat sea surface temperature as a long name after that (it happens in several other points in the manuscript)

(Figure 3) [not necessary, but suggested] I find that putting a single Figure only to show the regions studied is a quite unhappy choice. To economise I should suggest to plot a field of interest for the presentation (e.g. average SST anomaly, at your preference) and overplot the chosen basins. Also, I wouldn't talk about a subdivision since you are not considering the whole European Mediterranean subdivided in regions, instead you are choosing specific areas of interest.

Data/Methods

Change title in "Data and Methods"

(75) I believe that your Appendix B could be very much interesting to be included in motivation of taking the atmospheric variables that you analyse. I believe that introducing the upper-ocean energy balance by reporting solely Equation B1 is quite abrupt. Instead discussing more about it here or in introduction section would give a more robust background to your study choices. It is important at this level to know which atmospheric variables have been used throughout your work.

(76-78) not very interesting nor informative. Also, here you state that you use monthly data to compute the 1991-2020 climatology, then at (112) you say you use daily averages for climatology for the period 1982-2011. You should be clearer and consistent in the presentation.

(subsection 3.2.1) This section is very detailed. If the purpose of your work is to demonstrate the ability of NRT SST to capture response giving all this information could be justified. At the present version of the manuscript it sounds too detailed.

(subsection 3.2.2) it is misleading to entitle the section "SST analysis", as this sounds like you will give some results of your analysis already. You should opt for entitling it as "ESA CCI SST product" or "Satellite derived SST". Indeed, you have section 3.2.3 devoted to describe climatology computation, here you should introduce the product only

(122) replace "calculating" with "calculated". Replace "long-term value one" with "long-term one"

Results

(125) remove "use by the"

(125-128) This description is badly written. You talk about ideal case without saying what this means (that is complete data coverage). Please revise accordingly. Also, what do you mean with the basins missing data in percentage? Over the whole period considered? Please specify.

(136) take off "that affected these basins". Summer 2022 anomalies have not interested only your chosen basins.

(Figure 4) Revise date labels regarding August and make them consistent with the others (08-10 instead of 08-09 and so on). Put titles regarding the basins on each panel. Could be informative to report the percentage of missing data somewhere in panels.

(140) Revise how you write dates. For example, "between the 6th of July and…" should be "between July 6th…", similarly onwards.

(144) where are these anomalies shown?

(147) where is this given? Not evident to reader. Need to properly refer to figures.

(152) you report that temperatures are constantly above climatology by referring to Table, where only average values are given. State better

(Table 4 & Table A1). Table 4 does not exist, but only Table 1. I believe that confusion is made while describing content of tables. In text you say that Table 4 refers to July 31st -August 13th event, but caption of Table in text says differently. Revise accordingly. Consider to put them both in main text, to aid comparison and enable following values reported in text better.

(158) where do you show the trend?? You are showing average values in Figure 5.

(Figures 5 & 6) It could be useful to recall the regions selected overplotted on fields shown. Also, consider in showing only one of the two periods to not sound repetitive.

(165-171) where are these results shown in figures? Are you talking about Figure 6 23rd – 30th July event or August event? It is not easy to follow as a reader.

(Section 4.3) in my opinion contains very interesting comments. Consider to enlarge this description, deepening further the phenomenon and recalling it clearly in conclusive remarks.

(174) typo: *persistence

(179) specify that the anomaly you are talking about is of cloud cover

Discussion/Conclusions

Change title in "Discussion and Conclusion"

(198) typo: *studies

(202) limited? I think you meant the opposite

(204-206) What do you mean in terms of comparing other related studies to yours? It is not clear.

(200-206 & 213-244) The majority of concepts reported here sound more introductive than conclusive comments. I suggest to deeply revise Introduction & Discussions and Conclusion sections wisely.

(246) Not clear what you mean when you say "to prevent the detection of anomalously warm SST"

---

## Author Comment (AC1)

All of the co-authors would like to thank the referee for the time which he/she has allocated to the detailed revision of this paper, taking the time and effort to provide us with generally positive and constructive feedback. and the well-supported comments about our work. We sincerely value the work done in this review and we are grateful for this. We hope our responses and the improved version of the manuscript will meet the expectations.

Please, find below our point-to-point response with comments of the referee in **black** and our response in **blue**.

General comments

The authors of this study thank the reviewer for the general comments, which have highlighted numerous inaccuracies in our article. The main concern was the lack of a clear objective, which is evident upon rereading the article and has been noted by all reviewers. Initially, it seemed interesting to study both the ability of operational satellites to track oceanic extremes and the response of French coastlines to the exceptional summer of 2022. However, the study was conducted with the goal of characterizing the response of SSTs to multiple heat waves.
Therefore, the article has been revised to focus on this objective and determine the contribution of various atmospheric variables to this response. To achieve this, the structure of the paper has been revised to include necessary information for a more comprehensive study of ocean-atmosphere heat exchange interactions. Thereby, we conducted an analysis of the contribution of several atmospheric variables using a modelling approach. Hence, Materials & Methods, Results and Discussions have been largely modified.
In order to not start from scratch on the analysis of temperature data, we have kept the OSI SAF data and added details on comparison with the products used for climatology construction.

Minor comments

Across the manuscript, please clearly state how do you calculate anomalies. Are you comparing daily OSI/SAF SST values with monthly climatology? I couldn't get the point.
As mentioned by the reviewer, it is unclear how we calculated our anomalies. We, thus, provided more details about the process in the corresponding paragraph (section 2.3.3 on page 5).
We computed daily and monthly anomalies. For each, we compared daily (resp. monthly) ESA CCI product to the corresponding daily (resp. monthly) OSI SAF product. Thus we did not compare SST value that are on different time period.

Figures are too small. Please, make bigger figures.
Figures have been updated to be more readable.

Other comments

Lines 45-54 should not be included in the Introduction Section. A "Synoptic description/atmospheric characterization/..." section is needed and should go into greater detail.
A specific section has been added to describe the synoptic conditions.
P8, L.232 to P10, L250

Figures 1-2: I assume they are monthly anomalies although not stated. Maps are too small.

Figures have been modified. In fact, the anomalies shown are comparison of each heatwaves periods to the corresponding one of the 1991-2020 climatology.

Figure 3: Please, improve caption by stating the subdivisions are stated for the SST analysis

As suggested by the Reviewer.2 we decided to remove this Figure and to display the sub-areas in Figure 4. We included the following sentence in the legend:
"The English Channel (EC), the Bay of Biscaye (BB) and the North-western Mediterranean Sea (NWM) used in to analyse SST pattern throughout the 2022 meteorological summer are plotted on the subplot (a)."

Lines 69-71. Please, rewrite this sentence to improve the English language (missing verbs, concordance,...).

P3, L83-85.
"This area is directly influenced by Mistral and Tramontane regional winds which drive recurrent upwelling phenomena, making it of particular interest in comprehensive studies of the Mediterranean water cycle and its implications for climate studies (Drobinski et al., 2014; Ruti et al., 2016)"

Lines 76-77: Change "Both hourly and monthly were used" to "Both hourly and monthly data/values were used" (general remark: please, carefully check English grammar and spelling)

We finally removed this sentence as we included a specific paragraph to detail the construction of anomalies.

Line 142: Please change "3°CC"
Corrections made

Line 153: Reference to "Table 4" but this one does not exist. Maybe A1?
P13, L284. Correction has been made. There was a mismatch between references, here we intended to refer to the Figure 3.

Lines 160-164: Do these sentences refer to single point values? The 7.9ºC anomaly, is referred
to local or mean climatology? This is not clear for me. If this is compared to mean areal climatology this would not be the right way to compare daily values to climatology

This sentence aims to focalise on a single point and give further insights into the range of response and extremes values reached during the event that are hidden when analysing mean values.
However, the anomaly in question referred to the analysis of the local measured SSTs to the single point 1982-2011 climatology.
We propose the following corrected sentences:

P13, L294 – P13, L300.
"We investigated the local response to the marine heatwave in each basin by calculating the 1982-2011 daily climatology for every single point within each region. Our analysis revealed that the maximum recorded temperature was 30.8°C on August 4th

in the NWM area, 23.6°C on August 12th in the EC area, and 26.4°C on August 11th in the BB area. In terms of anomalies, the NWM basin exhibited the minimal anomaly of 2.2°C, whereas the EC and BB basins exhibited negative anomalies of -1.5°C and -2.1°C, respectively. The maximum anomalies were 7.9°C in NWM, 3°C in EC, and 3°C in BB, indicating the extensive response of the NWM basin and the range of sea surface temperature (SST) variability within and between each basin. The stronger SST variability in both BB and EC regions was notable."

Line 179: Which is the "whole domain"?
The whole domain refers to France.
P14, L319 – P14, L321.
"The mean total cloud cover anomaly over France reaches -17% while the North of France and specifically Brittany have undergone the maximum average anomaly of -37%"

Figure 7: Not stated but I understand that the maps show anomalies for the days in August 22 heatwave days, respect to 1991-2020 august monthly values? To the same period (31/07-13/08) for the 1991-2020? Please add this information in the figure caption and text.
We calculated the anomalies in respect to the corresponding period of the 1991-2020 climatology. This precision has been added in the figure caption and text.

Legend of Figure 5 on P.16
"Anomalies are compared to the same period in the 1991-2020 climatology."

Line 209: in response to the atmospheric heatwaves that affected France during the 2022 summer. Results and analysis are mostly centred in the August event, maybe not valid for the rest of summer.

Even if we centered part of our study on the August event, we also analysed the signal over the summer months.
We proposed the following sentence:

P22, L435-438. "Despite the significant lack of data, particularly in the early summer and in the EC area, we found a clear warming signal of SSTs during the summer of 2022 that was evident in all studied areas. All three areas exhibited positive SST anomalies throughout the summer, with record-breaking daily anomalies indicating that 2022 was one of the warmest summers in terms of SSTs, which also started early in the season."

Lines 209-213: "The strongest response was found 210 on the NWM basin (with a maximum average SST anomaly of 4.3°C) which is in line with observations (Bensoussan et al., 2019) and modeled evolution (Darmaraki et al., 2019b) confirming the Mediterranean Sea is a "hotspot" for climate change (Giorgi, 2006)". A single extreme event does not confirm the hotspot, although I agree with the sense of your assumption and that it is in line with the cited references. Please, rewrite this sentence.

We agree on this comment and on the fact that a single event does not prove that the Mediterranean Sea is a climate hotspot. To get more in line with the references and detail our thoughts we proposed this corrected sentence:

P.22, L449 – 455.
"The Mediterranean Sea is recognized as a "hotspot" for climate change (Giorgi, 2006), which will face warmer summer seasons (Adloff et al., 2015). Our findings support the idea that the occurrence of heatwaves throughout the summer would cause the

NWM Sea to respond strongly to these atmospheric forcings. Indeed, results indicate that even during non-heatwave periods, the SSTs in the NWM area were consistently warmer than the climatological average, even when the net heat flux was close to normal. These results are also in line with the with observations (Bensoussan et al., 2019) and modeled evolution (Darmaraki et al., 2019) of the continuous warming of the Mediterranean Sea."

Lines 227-244: These lines are not a discussion/conclusion but should be part of the introduction section. No work has been done regarding MHWs across the manuscript

We completely agree with this remark. In line with the general comment of Reviewer 1 and to a comment of Reviewer 2 that had the same questioning we decided to completely revised the Discussions/Conclusions section. The section integrates previous discussions with the addition of new features that are linked to the attribution results. The section starts P21, L429.

L246 : "To prevent the detection of anomalously warm SSTs". To prevent?
We propose the revised sentence:
P24, L506. « To anticipate the detection of anomalously warm SSTs »

Lines 250-256: Rephrase to improve reader understanding
As the section has been completed revised, this sentence is no longer present.

---

## Author Comment (AC2)

All of the co-authors would like to thank the referee for the time which he/she has allocated to the detailed revision of this paper, taking the time and effort to provide us with generally positive and constructive feedback. and the well-supported comments about our work. We sincerely value the work done in this review and we are grateful for this. We hope our responses and the improved version of the manuscript will meet the expectations.

Please, find below our point-to-point response with comments of the referee in **black** and our response in **blue**.

**General comments**

The authors of this study thank the reviewer for the general comments, which have highlighted numerous inaccuracies in our article. As stated in response to Reviewer 1, our main concern is the lack of a clear objective, which is evident upon rereading the article and has been noted by all reviewers.
Initially, it seemed interesting to study both the ability of operational satellites to track oceanic extremes and the response of French coastlines to the exceptional summer of 2022. However, the study was conducted with the goal of characterizing the response of oceans to multiple heatwaves.
Therefore, the article has been revised to focus on this objective and determine the contribution of various atmospheric variables to this response, allowing for a more seamless reading experience. To achieve this, the structure of the paper has been revised to include necessary information for a more comprehensive study of ocean-atmosphere heat exchange interactions. Thereby, we conducted an analysis of the contribution of several atmospheric variables using a modelling approach. Hence, Materials & Methods, Results and Discussions have been largely modified.
In order to not start from scratch on the analysis of temperature data, we have kept the OSI SAF data.

On the specific point about using two different SST data. As explained before, we started by analysing the response of the SST in an operational framework which explained why we kept these data in our study. In any case, both measurements are comparable as they reflect the nighttime SST (or corrected by the diurnal cycle on the specific case of ESA CCI CDR).
We added in the text an explanation on how they are correlated.

**Line by line comments**

Abstract

(1-3) These initial sentences should be more precise. From the beginning of which measurements? The seasonal average of 22.7°C is intended as surface air temperature, or what? Multiple record-breaking heatwaves over which period?

This sentence has been modified as following :

P1. L1-3
« Summer 2022 was memorable and record-breaking, ranking as the second hottest summer in France since 1900, with a seasonal surface air temperature average of 22.7°C. In particular, France experienced multiple record-breaking heat waves during the meteorological summer. »

(9-10) I would not introduce the core of your study saying that "the contribution of other atmospheric variables is not negligible". This sounds like an a-priori consideration. I believe, instead, that the choices of variables to investigate have a clear motivation and should be better introduced and made more attractive in the Abstract.

The sentence has been removed and replaced by :
P1. L6-7
« Beyond the direct relation between sea surface temperatures and the surface air temperatures, we explored the leading driving factors affecting the upper-layer ocean heat budget and determined the magnitude of such atmospheric factors. »

Introduction

(Figures 1 & 2) The images should be larger, could aid having titles referring to the months. How did you compute the 1991-2020 climatology to which you define anomalies? Since the main rationale of your work in exactly the anomaly in surface air temperature, consider if to switch order of Figures 1 & 2, or if to add a single Figure before these showing the mean spatial anomaly over whole summer, or if a time series.

Fig 1 & 2 have been modified by adding months in the titles. Order has been switched.
We thought about the proposition of adding a single Figure either as a map or a time series. In our opinion, two figures give complementary information about the atmospheric circulation over the summer and the consequences on the surface air temperature.

(25-32) You longly talk about OHC and the reader it brought to think he/she will see some analysis on OHC, which instead is not present. I suggest you shorten OHC description, or you add an analysis of the field in summer 2022.

We completely agree on this comment, thus this section has been removed as it is not related to the topic of this study.

(45-48) As presented this part sounds as if you are describing already some results, because you go specific on dates and causality of events with no reference. I suggest you keep introduction to be more descriptive, eventually recalling in results more specific relations of causation.
Thanks for this valuable comment which we completely agree on. As proposed by both reviewers we added a specific section describing in details the synoptic condition.
Paragraph starting on P8, L232.

Study sites

(64) Since you introduce the SST acronym at line 37, you should not repeat sea surface temperature as a long name after that (it happens in several other points in the manuscript)

P3, L78. This has been modified.

(Figure 3) [not necessary, but suggested] I find that putting a single Figure only to show the regions studied is a quite unhappy choice. To economise I should suggest to plot a field of interest for the presentation (e.g. average SST anomaly, at your preference) and

overplot the chosen basins. Also, I wouldn't talk about a subdivision since you are not considering the whole European Mediterranean subdivided in regions, instead you are choosing specific areas of interest.

As proposed we removed the Figure 3 and overplotted studied areas on the Figure 4 (P15).

Data/Methods

Title has been changed

(76-78) not very interesting nor informative. Also, here you state that you use monthly data to compute the 1991-2020 climatology, then at (112) you say you use daily averages for climatology for the period 1982-2011. You should be clearer and consistent in the presentation.
We agree with the reviewer about the inconsistencies throughout the paper about how we calculated the anomalies. In the present version, we made the calculation procedure clearer and we dissociated the description of the atmospheric climatology from the oceanic one.
To improve the readability of the anomalies computation we decided to have a distinct section (P5, L122 – L133).

(subsection 3.2.1) This section is very detailed. If the purpose of your work is to demonstrate the ability of NRT SST to capture response giving all this information could be justified. At the present version of the manuscript it sounds too detailed.
P4, L97-L113. This section has been shortened in order to be consistent with the objective of this study.

(subsection 3.2.2) it is misleading to entitle the section "SST analysis", as this sounds like you will give some results of your analysis already. You should opt for entitling it as "ESA CCI SST product" or "Satellite derived SST". Indeed, you have section 3.2.3 devoted to describe climatology computation, here you should introduce the product only.
P4, L114. We changed this subsection to detail the ESA CCI product only.

(122) replace "calculating" with "calculated". Replace "long-term value one" with "long-term one"
As we modified the subsection 'SST climatology and anomaly », this sentence no longer exists.

Results

L125 remove "use by the"
P11, L253.
"The primary objective of an operational product is to provide daily monitoring for use by forecasting services."

(125-128) This description is badly written. You talk about ideal case without saying what this means (that is complete data coverage). Please revise accordingly. Also, what do you mean with the basins missing data in percentage? Over the whole period considered? Please specify.

This section has been improved to improve the readability. A reference to the section detailing what are the missing data has been also added.

P6, L125-128.
"The first objective of an operational product is to provide a daily usable monitoring for use by the forecasting services. However, these conditions are ideal cases and are therefore not met in all basins every day. Thus a significant part of the data is not available depending on multivariate conditions (clouds, aerosols, low quality data). This share varies for each basin and is compiled in the Figure 4."

(136) take off "that affected these basins". Summer 2022 anomalies have not interested only your chosen basins.
Corrections made

(Figure 4) Revise date labels regarding August and make them consistent with the others (08-10 instead of 08-09 and so on). Put titles regarding the basins on each panel. Could be informative to report the percentage of missing data somewhere in panels.
Figure 4 has been improved thanks to your comment.

(140) Revise how you write dates. For example, "between the 6th of July and..." should be "between July 6th...", similarly onwards
We have checked all the date to have a consistent and standardised format

L144 where are these anomalies shown?
Reference to the table 2 was missing thus we added it in the sentence. We also modified the table 2 to improve readability. We also added the variation of coefficient in addition to the standard deviation to get more insights in the variability of the SST anomalies.

P11, L274-276.
« This is also high- lighted by the variability, presented in the Table 2, which is comprised between 31% and 46% of the mean SST anomaly for a standard deviation between 0.5°C and 0.8°C »

L147 where is this given? Not evident to reader. Need to properly refer to figures.
We finally decided to remove this sentence and modify it to the following one:

P11, L276-279.
"With the exception of specific episodes, SSTs remain close to the climatological maximum of the period 1982–2011 (Fig.3). In addition, it is noteworthy that the NWM experienced 22 days, EC experienced 19 days, and BB experienced 4 days of SSTs exceeding the climatological maximum. It should be noted that the previous temperature record in the NWM dated back to 2003, underscoring the historical significance of the observed response."

(152) you report that temperatures are constantly above climatology by referring to Table, where only average values are given. State better
Reference to the Figure instead of the Table has been done

P12-13, L283-284. « During this period, SSTs were abnormally high, with temperatures constantly above the climatological norm (as shown in the Fig 3). »

(Table 4 & Table A1). Table 4 does not exist, but only Table 1. I believe that confusion is made while describing content of tables. In text you say that Table 4 refers to July 31st - August 13th event, but caption of Table in text says differently. Revise accordingly. Consider to put them both in main text, to aid comparison and enable following values reported in text better.

As proposed by the reviewer, we put the table A1 in the main text (it is now referenced as Table 2).

The mentioned confusion comes from the mismatch between Table and Figure. We have corrected this by referring to the correct reference (Figure 3 instead of Table 4). We have revised the figure/table captions and the mentioned referenced in the text.

(158) where do you show the trend?? You are showing average values in Figure 5.

This is a mistake in the text, actually we were not intended to talk about a trend. Our point is to show that this warming is uniform and affect all the studied areas as seen in Figure 4.

P13, L290-291.

«Positive temperature anomalies were found throughout the majority of the ocean surface and the trend of increase was spatially uniform (Fig. 4) »

(Figures 5 & 6) It could be useful to recall the regions selected overplotted on fields shown. Also, consider in showing only one of the two periods to not sound repetitive.

As proposed by the reviewer, we consider that adding the studied regions on this figure will help readers to follow the study. We also rearranged the figures to finally keep only one representing better the focus of our study.

(165-171) where are these results shown in figures? Are you talking about Figure 6 23rd – 30th July event or August event? It is not easy to follow as a reader.

This paragraph referred to the comparison between the Figure 4 and 5 related to the July 31th-August 13th heatwave. We add a precision in the text and make the reference to the correct figure.

« As previously mentioned in Section 5,1, SSTs were already abnormally warm before the August 31th to August 13th heatwaves. »

(Section 4.3) in my opinion contains very interesting comments. Consider to enlarge this description, deepening further the phenomenon and recalling it clearly in conclusive remarks.

Consequently to this comment and a similar one from Reviewer 1 we decided to get further insights into the contribution of atmospheric variables by adding further developments. To address this we conducted an attribution analysis based on a modelling approach. The results are presented in the section 4.4 starting on P16, L338.

(174) typo: *persistence
Typo corrected

(179) specify that the anomaly you are talking about is of cloud cover
P14, L319-321.
"The mean total cloud cover anomaly over France reaches -17% while the North of France and specifically Brittany have undergone the maximum average anomaly of -37%"

Discussions

(198) typo: *studies
Typo corrected.

(246) Not clear what you mean when you say "to prevent the detection of anomalously warm SST"
P24, L506 « To anticipate the detection of anomalously warm SSTs »

Comments of the lines 204-206, 200-206 & 213/244

In addition to the newly introduced features that were added to the results section to reflect the sensitivity test, we have also reorganized and rephrased the Discussion/Conclusion section. Specifically, we incorporated the feedback provided by Reviewer 2 and removed all introductory comments to streamline the content.

---

## Referee Report (RR1)

**Title: Response of the sea surface temperature to heatwaves during the France 2022 meteorological summer**

**Author(s): Thibault Guinaldo, Aurore Voldoire, Robin Waldman, Stéphane Saux Picart, and Hervé Roquet MS No.: egusphere-2022-1119**

**Minor comments**

**Abstract**

- *"Summer 2022 was a memorable and record-breaking event, ranking as the second hottest summer in France since 1900, with a seasonal surface air temperature average of 22.7°C"*: This is not an event, it is a whole summer/season.
- *"SSTs beyond the climatological maximum"*. What is a climatological maximum?
- "*Our results are in line with previous studies, and demonstrate that even if the 20 Mediterranean is known as a climate change hotspot, all the studied maritime areas are affected by a continuous warming of surface water and responded to extreme synoptic conditions."* Unclear sentence, please rewrite. I can't see the connection between Med being a hotspot and the SST warming and the response to synoptic conditions.

**Introduction**

- *"However, the operational and climatic needs for observations require a spatial and temporal coverage that cannot be achieved by in situ measurements alone"*. I can't get the meaning/context of this sentence in the paragraph.

**Data and methods**

**Study sites**

- Change *"we focus on the response"* to *"we focus on SST response"*
- *"as presented in Figure 4."*. Usually this type of information is presented in figure 1 to let the reader have a clear idea of the study area/domain before any data analysis or consideration. Figure numbers must follow a numbering correlative to their appearance in the text. It should be easy and possible to add the domains to actual fig 1.

**Atmospheric reanalysis**

- *"We also used an atmospheric forcing climatology to test the heat flux sensitivy"*. What does an atmospheric forcing climatology mean? From ERA5 data? Any other source.

**Sea surface temperature data**

- *"2.3.1 Operatio**nn**al SST product"*

**Daily sea surface temperature evolution over the 2022 meteorological summer**

- *In particular, the missing data does not permit a systematic analysis of the response of the three 260 basins to the early heatwave in June 2022 (June 15th to 19th) and the July 2022 (July 11th to 25th) heatwave. However, the response of the July 2022 heatwave is conceivable for the BB and NWM areas.* Is this not contradictory? Is it possible to address the July event or not? Please, state which basin is possible to assess in each HW.
- *"Notably, there were no days at the basin scale where temperatures were within the normal temperatures range or below."* I assume it is in the NWM.
- *"The summer of 2022 also a record for this basin with an average temperature of 26.1 °C."* A verb is missing.

**Observed variability of the atmospheric variables**

- *"As presented in the Figure 5a, we looked, in the first place"* should it be just *"figure 5"*?
- *"The daily anomaly during the period of the heatwave is significantly correlated to the anomaly of SSTs in the NWM area with a Spearman coefficient of 0.8"*. What about BB and EC? Additionally, what do you correlate? Mean surface solar radiation over whole France? Over NWM domain? It is not clearly stated.

**Conclusions**

- *The magnitude of anomalies can also be attributed to anthropogenic forcing, which can be quantified using singular event5 detection and attribution cutting-edge approaches (Ribes et al., 2020; Faranda et al., 2022).* Not a conclusion from the authors. In a single sentence paragraph?
- *Lines 522-531 are not discussion or conclusions*

**Other**

- You could supress Table 1. It does not add much information, model experiments are well described in the text.
- Figure 3. Having the same dimension/limits in the the y axis will make all plots comparable and illustrate the different intensities of the SST anomaly
- Figure 5. It is not necessary to repeat the same scales and titles in the x axis. I suggest moving the color legends to a vertical legend located at the right side of the plot and only use one x title at the bottom of each column. It will gain space, improve readability offering the same information.
- Figures 6, 7 and 8. Please, in the captions explain sequentially the different plots; (b) plot is not for SST, it should say (c). Uising the same colors. Do sensitivity tests relate to experiments in Table 1? Are line colors in plot (b) the same than in (d)? If there are colors there has to be a legend. If there is a common legend place it adequately. Maybe place the "sensitivity tests" plots as (c) and (d)
- Parenthesis missing in lines 389 and 390
- Line 395. If SSR is an acronym for "surface solar radiation" should have been introduced in the first appearance. I could not find before.

---

## Author Response (AR2)

Again all of the co-authors would like to thank the referee for the time which he/she has allocated to the detailed revision of this paper and the constructive comments. We hope our responses and the improved version of the manuscript will meet the expectations.

Please, find below our point-to-point response with comments of the referee in **black** and our response in **blue**.

**Abstract**

"Summer 2022 was a memorable and record-breaking event, ranking as the second hottest summer in France since 1900, with a seasonal surface air temperature average of 22.7°C": This is not an event, it is a whole summer/season.
"Summer 2022 was memorable and record-breaking, ranking as the second hottest summer in France since 1900, with a seasonal surface air temperature average of 22.7\unit{°C}."

"SSTs beyond the climatological maximum". What is a climatological maximum?
"The studied areas experienced between 4 and 22 days where the basin-averaged SSTs exceeded the maximum recorded basin-averaged SSTs from 1982 to 2011."

"Our results are in line with previous studies, and demonstrate that even if the 20 Mediterranean is known as a climate change hotspot, all the studied maritime areas are affected by a continuous warming of surface water and responded to extreme synoptic conditions." Unclear sentence, please rewrite. I can't see the connection between Med being a hotspot and the SST warming and the response to synoptic conditions.
"Our study findings are consistent with previous research and demonstrate the vulnerability of the Mediterranean Sea to the increasing frequency of extreme weather events resulting from climate change. Furthermore, our investigation reveals that the recurring heatwave episodes during the summer of 2022 had an undeniable impact on all the surveyed maritime areas in France."

**Introduction**

"However, the operational and climatic needs for observations require a spatial and temporal coverage that cannot be achieved by in situ measurements alone".
Here we justify the use of satellite data and the fact that these data play a key role in responding to the operational forecasting and climatic projections needs.

**Data & Methods**

**Study Sites**

All of the reviewers comments have been addressed.

**Atmospheric reanalysis**

"We also used an atmospheric forcing climatology to test the heat flux sensititivy". What does an atmospheric forcing climatology mean? From ERA5 data? Any other source.
We agree with the reviewer that the sentence is unclear. We modified the sentence as follow:

"We also used ERA5 data as atmospheric forcing climatology to test the heat flux sensitivity the each variable in the model experiments."

**Sea surface temperature data**

The title has been modified.

**Daily sea surface temperature evolution over the 2022 meteorological summer**

In particular, the missing data does not permit a systematic analysis of the response of the three 260 basins to the early heatwave in June 2022 (June 15th to 19th) and the July 2022 (July 11th to 25th) heatwave. However, the response of the July 2022 heatwave is conceivable for the BB and NWM areas. Is this not contradictory? Is it possible to address the July event or not? Please, state which basin is possible to assess in each HW. Sentences have been modified to address the need for clarity.

"In particular, the missing data does not permit a systematic analysis of the response between the three basins to the early heatwave in June 2022 (June 15th to 19th) heatwave. The response of the July 2022 (July 11th to 25th)  heatwave is individually conceivable for the BB and NWM areas. Only the response to the August 2022 heatwave is feasible for all three basins. Therefore, the analysis of the SST response to heatwaves in this study focuses on the August 2022 event."

"Notably, there were no days at the basin scale where temperatures were within the normal temperatures range or below." I assume it is in the NWM.

"Notably, there were no days at the NWM basin scale where temperatures were within the normal temperatures range or below."

"The summer of 2022 also a record for this basin with an average temperature of 26.1°C." A verb is missing.
Verb has been added.
"The summer of 2022 also set a record for this basin with an average temperature of 26.1°C."

**Observed variability of the atmospheric variables**

"As presented in the Figure 5a, we looked, in the first place" should it be just "figure 5"? The reference to the Fig5a has been modified to Figure 5.

"The daily anomaly during the period of the heatwave is significantly correlated to the anomaly of SSTs in the NWM area with a Spearman coefficient of 0.8". What about BB and EC? Additionally, what do you correlate? Mean surface solar radiation over whole France? Over NWM domain? It is not clearly stated.

Sentence has been modified as follow:

"The daily anomaly over the NWM domain during the period of the heatwave is significantly correlated to the anomaly of SSTs in the area with a Spearman coefficient of 0.8"

**Conclusions**

The magnitude of anomalies can also be attributed to anthropogenic forcing, which can be quantified using singular event5 detection and attribution cutting-edge approaches (Ribes et al., 2020; Faranda et al., 2022). Not a conclusion from the authors. In a single sentence paragraph?

Lines 522-531 are not discussion or conclusions

The page break is due to formatting caused by the end of the page in the PDF. This has been corrected.
In the context of our study, and as it represents a starting point for analyzing SSTs during the summer of 2022, we wanted to construct our discussion and conclusion section by including perspectives on future work that could address the limitations of our study while also anticipating various factors that could improve and enrich our conclusions. This explains the discussions on the attribution to anthropogenic forcing as well as lines 522-531.

**Other**

You could supress Table 1. It does not add much information, model experiments are well described in the text.
Table removed.

Figure 3. Having the same dimension/limits in the the y axis will make all plots comparable and illustrate the different intensities of the SST anomaly.
We understand the point raised by the reviewer, however we prefer to keep the figure as it is in order to help the analysis of the response of each basin. Having the same limits will result in a shrinkage of the EC and BB SSTs which will, in our opinion, reduce the readability of the figures.

Figure 5. It is not necessary to repeat the same scales and titles in the x axis. I suggest moving the color legends to a vertical legend located at the right side of the plot and only use one x title at the bottom of each column. It will gain space, improve readability offering the same information.
We assume that this comment refers to Figure 4 rather than Figure 5. In this case, we have taken the comments into account, but we still prefer to keep a horizontal colorbar at the bottom of each column.

Figures 6, 7 and 8.
Figures captions have been corrected.

Parenthesis missing in lines 389 and 390.
Corrected.

Line 395. If SSR is an acronym for "surface solar radiation" should have been introduced in the first appearance. I could not find before.

The use of SSR was a typo, we defined the acronym for surface solar radiation on the line 222. However, we prefer here to use 'surface solar radiation' here to increase readability.